# Late Holocene fast-ice dynamics around the Northern Victoria Land coast, Antarctica

T. Tesi[1] ✉, M. E. Weber[2], F. Muschitiello[3,4], D. Dutta[3,5], S. T. Belt[6], C. Pambianco[1,7], A. Di Roberto[8], L. Silva[9], K. Gariboldi[9], C. Morigi[8], F. Battaglia[1], E. Colizza[10], L. De Santis[11], A. Gallerani[12], G. Aulicino[1,13], L. Langone[1] & P. Giordano[1]

Fast ice, a consolidated form of sea ice commonly found along the Antarctic margins, plays a critical and multifaceted role in regulating ocean–cryosphere interactions and ecosystem dynamics. While satellite observations and numerical models provide valuable contemporary insights, reconstructing long-term trends and identifying potential environmental drivers requires alternative approaches. In this study, we present a novel method for reconstructing long-term fast-ice dynamics using a high-resolution analysis of a laminated sedimentary record from Northern Victoria Land, Antarctica. By integrating biomarker data, diatom assemblages and image analysis at sub-millimeter scale, we show how laminated deposits accurately reflect fast-ice variability, offering a new tool to investigate periods beyond the reach of direct observation. Our 3700-year record reveals persistent low-frequency cyclic patterns aligned with known solar cycles (Gleissberg and De Vries), pointing to a possible link between solar variability and fast-ice breakup through perturbation of regional atmospheric forcing. These results demonstrate the potential of our approach to resolve past changes and analyze temporal patterns in fast ice behavior during the late Holocene.

Along the Antarctic margins, fast ice represents a form of highly consolidated sea ice anchored to the coastline and ice shelf fronts. Fast ice is also present in shallow waters where grounded icebergs reside for long periods[1–3]. Fast ice typically forms long bands up to 200 km wide, of annual to multi-year age, with some regions experiencing cyclical breakouts and regrowing[2,4,5]. Antarctic fast ice plays a pivotal role in ocean-cryosphere interactions and ecosystem functioning[6]. For instance, its distribution influences the size of coastal polynyas[7], impacts on regional sea-ice production rates and facilitates the intrusion of modified Circumpolar Deep Water into the ice shelf[8] and the formation of Antarctic bottom water[9]. Recent studies highlight the mechanical role of fast ice in bonding and stabilizing outer margins of floating glacier tongues and ice shelves[10]. In addition, fast ice serves as a vital breeding habitat for emperor

[1]Istituto di Scienze Polari - Consiglio Nazionale delle Ricerche ISP-CNR, Bologna, Italy. [2]Institute for Geosciences, Department of Geochemistry and Petrology, University of Bonn, Bonn, Germany. [3]Department of Geography, University of Cambridge, Cambridge, UK. [4]Centre for Climate Repair at Cambridge, Department of Applied Mathematics and Theoretical Physics, University of Cambridge, Cambridge, UK. [5]Research School of Earth Sciences, Australian National University, Canberra, ACT, Australia. [6]Biogeochemistry Research Centre, School of Geography, Earth and Environmental Sciences, University of Plymouth, Drake Circus, Plymouth, Devon, UK. [7]Campus Madonna delle Piane, Università degli Studi "G. d'Annunzio" Chieti-Pescara, Chieti (Chieti Scalo, Italy. [8]Istituto Nazionale di Geofisica e Vulcanologia (INGV), Sezione di Pisa, Pisa, Italy. [9]Department of Earth Sciences, University of Pisa, Pisa, Italy. [10]Dipartimento di Matematica, Informatica e Geoscienze, Università di Trieste, Trieste, Italy. [11]Istituto Nazionale di Oceanografia e di Geofisica Sperimentale, OGS, Sgonico (Trieste), Italy. [12]Istituto di Scienze Marine, Consiglio Nazionale delle Ricerche ISMAR-CNR, Bologna, Italy. [13]Dipartimento di Scienze e Tecnologie, Università degli Studi di Napoli "Parthenope", Napoli, Italy. ✉e-mail: tommaso.tesi@cnr.it

penguins[11] and seal foraging[12], structures shallow coastal benthic ecosystems, and fosters high primary productivity, particularly through concentrated sea ice algal growth[13]. Furthermore, fast ice acts as a reservoir of nutrients, enhancing coastal zone primary production upon melt[14]. Lastly, the presence of fast ice permits aviation in some Antarctic regions, thus supporting resupply activities over summer[15].

A wide range of different possible mechanisms has been identified to explain fast-ice variability including atmospheric forcing, break-up driven by swell, anomalous snow cover, basal melt driven by warm water mass intrusion and bathymetric control[1,6]. However, an overarching and dominant mechanism has not emerged and thus, resolving fast ice temporal variability and its relationship with climate remains largely unknown. Although models and satellite images have been used to investigate temporal trends[1,3,16,17], fast ice modelling is still in its infancy since most models are either unable to simulate fast ice (none of the models contributing to Coupled Model Intercomparison Project Phase 6 (CMIP6) incorporates fast ice, explicitly[18]) or it is highly regional[19,20]. In this respect, satellite image analysis is far more accurate. For instance, Fraser et al. (2020) used NASA Moderate Resolution Imaging Spectroradiometer (MODIS) data to produce the first comprehensive circum-Antarctic time-series of fast-ice distribution. This study covers 18-y of acquisition and reveals statistically significant trends, with Bellingshausen Sea (60–102°W) exhibiting the largest positive trend (about 2.8% y⁻¹) while the largest negative trends were observed in Weddell Sea (27–60°W) (about −2.59% y⁻¹). However, despite their quantitative capacity at a pan-Antarctic scale, observations through satellites are limited by the time constraint (i.e., only a few decades) that hampers our understanding of longer-term natural cycles. Such knowledge is critical, however, to distinguish the complexity of natural climate variability from human-induced changes.

In this study, we show-case a novel approach to reconstruct long-term fast-ice dynamics (i.e., beyond instrumental satellite measurements) by analysing marine sediments collected from Northern Victoria Land spanning the last 3.7 ka to resolve multi-decadal and centennial-scale variability (Fig. 1). Today, the Northern Victoria Land region is characterized by a fast ice of about $2.6 \pm 1.0$ m thickness[21]. We focused on a well-laminated sediment core (TR17-08; Fig. 1) collected in Edisto Inlet (Ross Sea) and used lamination intensity and frequency as a proxy of fast-ice variability for the Northern Victoria Land region.

Laminated sediments have been observed in various depositional settings around Antarctica, where they provide high-resolution records of environmental and climatic variability. In the Dumont d'Urville Trough (East Antarctica), Maddison et al.[22] documented seasonal to sub-seasonal laminae, recording diatom bloom cycles driven by changes in sea ice, nutrient concentration and water column stability. Denis et al.[23] used laminated patterns in sediments from Adelie Land region (East Antarctica) as an indicators of changes in bottom water production and sedimentary processes. On the MacRobertson Shelf (East Antarctica), Alley et al.[24] described continuously laminated late Holocene sediments from Iceberg Alley, where mm- to cm-scale layers reflect high-frequency productivity pulses sustained by meltwater and nutrient input from sea ice and icebergs. Similarly, in the Palmer Deep, Leventer et al.[25] identified seasonal diatom laminations interpreted as depositional cycles tied to spring–summer productivity, offering a longer-term perspective on Holocene oceanographic variability.

Lamination are also common in the Ross Sea, especially inside inlets[26,27]. Following previous work in the Edisto Inlet[26] performed on a nearby core (HLF17-01; Fig. 1) we merged diatom assemblages and organic geochemical approaches to interpret the laminated patterns and performed grey-scale curve generation and automated laminae counting based on high-resolution image analysis[28]. By combining grey-scale image analysis with the taxonomic and chemical fingerprint of the laminated pattern, we were able to resolve the ecological responses to various fast-ice conditions at a sub-millimetre scale. As a result, we produced a high-resolution and continuous record of fast-ice variability covering the late Holocene and conducted time-frequency analyses of fast ice to discern multi-decadal and centennial-scale signals. We also incorporated one year of in-situ observations from the Edisto Inlet, along with MODIS satellite imagery, to better interpret and contextualize the sediment proxies. Finally, to investigate the Southern Hemisphere's surface ocean and atmospheric response to increased solar radiation, we turned to CMIP6 coupled chemistry–climate model experiments. Overall, our multi-faceted approach offers a new perspective into the role of solar forcing on fast-ice variability, providing a novel insight into the natural variability of the Antarctic cryosphere.

## Results

### Core TR17-08: geochronology and lamination pattern

The Edisto Inlet is located in the northwestern part of the Ross Sea near Cape Hallett (Fig. 1a) and is characterized by the presence of a seasonal fast ice cover[21] as well as an expanded Holocene sedimentary sequence up to 130 m thick, in the middle of the fjord[29]. In the inlet, piston core TR17-08 (14.5 meters; Fig. 2) was recovered at a water depth of 462 meters (72° 18.2778′ S, 170° 04.1784′ E). Sediments consist of soft diatomaceous ooze exhibiting a recurrent pattern of light laminae overlying a darker sediment background. An age-depth model was previously developed for TR17-08 by Di Roberto et al.[30] based on 10 radiocarbon dates using the Marine13 calibration curve and one tephra layer deposited during the Mountain Rittman eruption in 1254 A.D.[31]. We ran ²¹⁰Pb measurements that revealed that, however, the core is missing its top section (i.e., absence of excess ²¹⁰Pb). We also analysed a short core taken with a multicorer (LS23-MUC17-8) at the piston coring site that instead shows excess ²¹⁰Pb in the upper 14 cm (Figure S1). Therefore, the age-depth model of TR17-08 was estimated between the tephra horizon and the lowermost radiocarbon date for a more conservative approach (Figure S2). In the current study, we reassessed the age model using the more recent Marine20 curve. Further details about the age-depth model can be found in the Methods (Section 4.2) and Supplementary Material. Overall, the updated age model suggests that the record extends back in time to 3.7 ka BP, consistent with Di Roberto et al.[30] despite the different calibration curve.

Using the BMPix & PEAK tools[28], we determined the amount and thickness variability of laminations (Fig. 3; Fig. S3). For core TR17-08, the maximum count method shows 575 maxima, 592 minima, 603 transitions from dark to bright and 600 transitions from bright to dark, revealing a difference of 28 laminae or ~4.5 %. With sufficient spatial measurement resolution (0.5 mm), our analysis further reveals that both bright and dark intervals range in thickness from 2 mm to ~5 cm (Fig. 3b, c). Thus, based on the image analysis for laminae recognition and counting, we conclude that the laminations do not simply represent annual varves, as the number of intervals detected with the PEAK tool is significantly fewer than the age of the sediment record. This implies that the lamination pattern cannot only reflect the seasonal thawing cycle typically found in fast ice, thus, other processes operating on a longer timescale must be responsible for the lamination pattern.

### Origin of the laminated pattern

In order to understand the origin of the sedimentary laminations and to test their relationship with fast-ice dynamics, we subsampled a pair of dark and light sediments from each section of TR17-08 core following the same approach described by Tesi et al.[26] for the nearby HLF17-1 core (Fig. 1). We analysed $\delta^{13}C$ (organic material), the highly branched isoprenoid biomarker IPSO₂₅ and characterized the diatom assemblages for each dark and light sediment sample; subsequently, we merged these results with data by Tesi et al.[26] who focused on a 1 m long section of laminated sediments in HLF17-1, spanning about

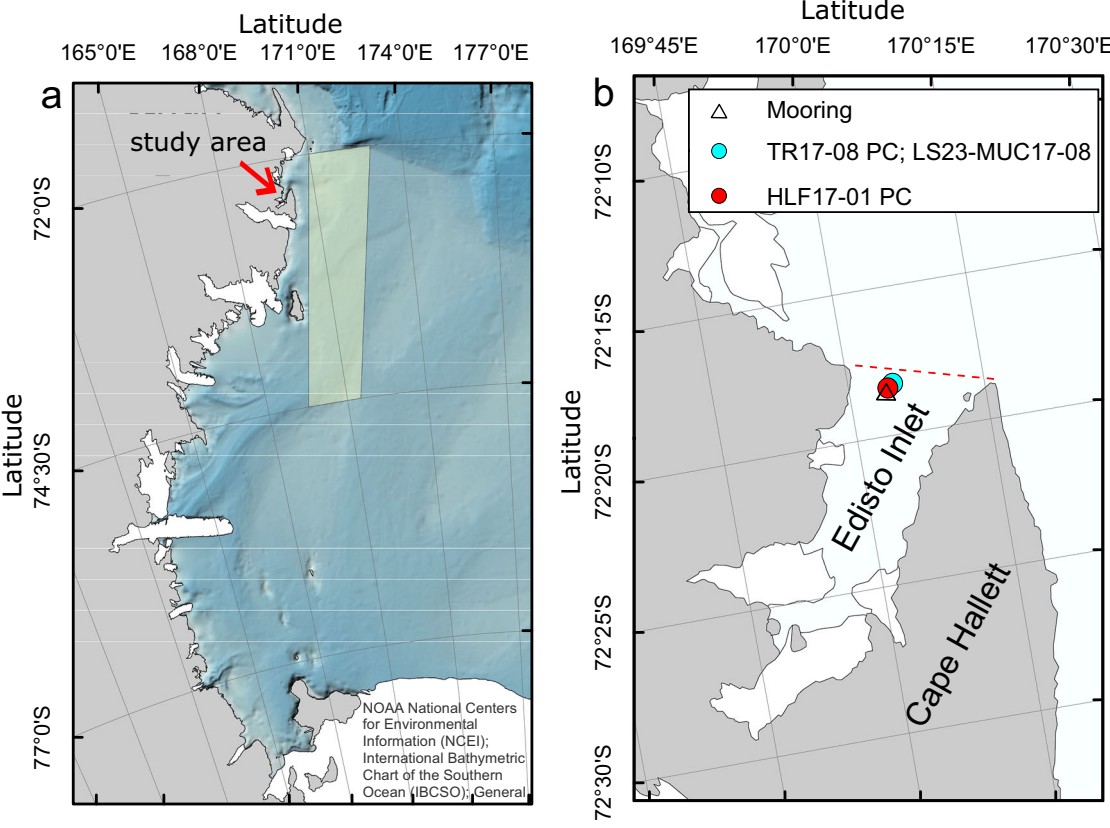

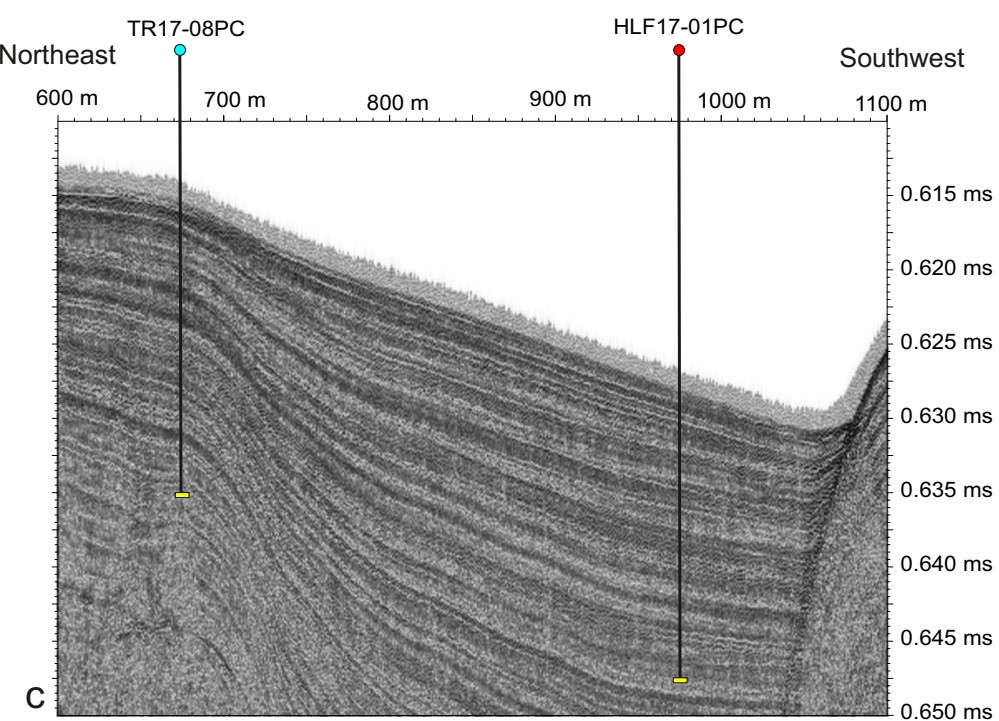

**Fig. 1 | Map of the study area in the Ross Sea. a** Victoria Land coast in the Ross Sea; the arrow shows the location of Edisto Inlet; the polygon displays the area where we assessed the concentration of pack ice (Fig. 7). **b** The orange dot shows the TR17-08 and LS23-MUC17-8 sampling site in Edisto Inlet; as a reference, the location of core HLF17-01 previously published and discussed in this study[26], is also shown; the dashed red line shows the limit of the inlet area where we calculated the percentage of the ice-free inlet (Fig. 8); the open star shows the location of the mooring line. **c** Sub-bottom profile along the coring sites showing the expanded, stratified sedimentary section in the Edisto Inlet[29].

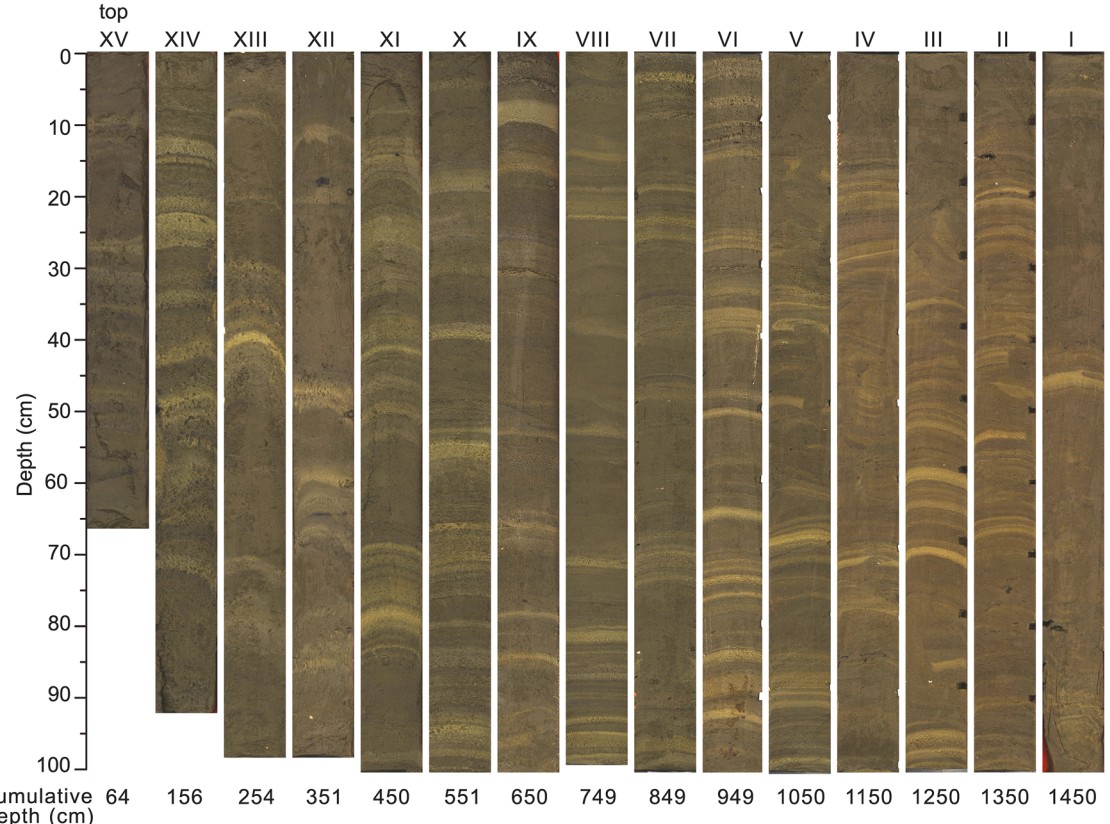

**Fig. 2 | Images of the TR17-08 piston core from top (left) to bottom (right).** Cumulative depth from the top core is shown at the bottom.

200 years (Fig. 4). Overall, the new geochemical and diatom data obtained from TR17-08 covering the last ~3.7 ka are consistent with these previous HLF17-1 results (Fig. 4a). We infer that the distinct compositional differences between dark and light sediment layers are indicative of contrasting fast-ice conditions. Dark intervals show a relatively higher contribution of sea ice-associated diatoms such as *Fragiliariopsis curta* and *F. obliquecostata* (Fig. 4b, c), are enriched in $\delta^{13}C$, consistent with an elevated contribution from sea ice algae[32] and contain the highest concentrations of $IPSO_{25}$, a biomarker made by *Berkeleya adeliensis*, a typical sympagic diatom found in fast ice (Fig. 4a)[33]. Overall, based on the sediment composition, we conclude that dark sediments record the early summer thawing when sympagic diatoms are released, while fast ice thawing stimulates the production of sea ice-associated diatoms. The larger variability observed in the dark sediments likely reflects changes in the relative contribution of individual sympagic and sea-ice-associated diatoms, rather than broad-scale environmental shifts.

In contrast, the light laminae are more uniform and primarily composed of the diatom *Corethron pennatum* (about 90%) (Fig. 4a). This shift towards semi-monospecific assemblages is indicative of the prolonged opening of the inlet after the initial breakup, possibly towards the end of summer when the water column is well-stratified due to the protracted fast ice thawing and freshwater input from local glaciers. *Corethron pennatum* is, in fact, well-adapted to compete in oligotrophic and stratified conditions through vertical migration within the water column to access nutrients below the pycnocline and evade competition with other algae[24]. Light laminae are also characterized by relatively low concentrations of $IPSO_{25}$, lower abundances of fast ice or sea ice-associated diatoms, and depleted $\delta^{13}C$ values, further supporting ice-free conditions (Fig. 4a).

To further corroborate our interpretation on the proposed temporal variability, we analysed sinking particles collected for 1-y (from March-2022 until February-2023) by a mooring line fitted with a sediment trap (Fig. 5). The mooring line was deployed close to the coring site (Fig. 1) at the same location where HLF17-01 core was retrieved. At the end of the Austral summer 2022 in ice-free conditions, both sediment and OC fluxes are relatively high while the $IPSO_{25}$ flux is negligible (i.e., no input from thawing fast ice) (Fig. 5b). During this period, $\delta^{13}C$ exhibits the most depleted values typical of open-water phytoplankton (Fig. 5a). In contrast, during the fast-ice breakup at the beginning of the Austral summer 2023, the flux of $IPSO_{25}$ increases substantially and $\delta^{13}C$ becomes relatively heavier (Fig. 5b). The seasonal shifts provide further support to our geochemical fingerprinting of laminated sediments in terms of contrasting fast-ice dynamics and, thus, our proxies are useful diagnostic tools for reconstructing different past fast-ice conditions in sediment archives.

Finally, the image analysis indicates that the overall contribution of bright and dark layers is ~1:1 (Fig. 3b, c). However, bright layers appear visually more distinct than dark layers (Fig. 2). This can also be derived from the normalized brightness (Fig. 3a), where bright layers represent frequent, positive peaks between 1.2 and 1.8, whereas the darker zones form more of a base level between 0.75 and -1. This likely implies that open-water terminated rather abruptly, whereas less severe fast ice/sea-ice conditions occurred for the remainder of the time. Given the overall count of ~600 layers in ~3 ka, this further indicates that the sharp bright layers represent seasonal, short-lived open-water conditions, whereas the rather diffuse dark layers represent multiple years of accumulation under fast ice coverage. This interpretation aligns with the diatom assemblages, where white laminae are primarily composed of *C. pennatum*, while dark sediments are dominated by sea ice and sea ice-associated diatoms (Fig. 4a) as previously discussed. To highlight the high variability of fast ice dynamics in the inlet which ultimately drives the laminated pattern and colour of TR17-

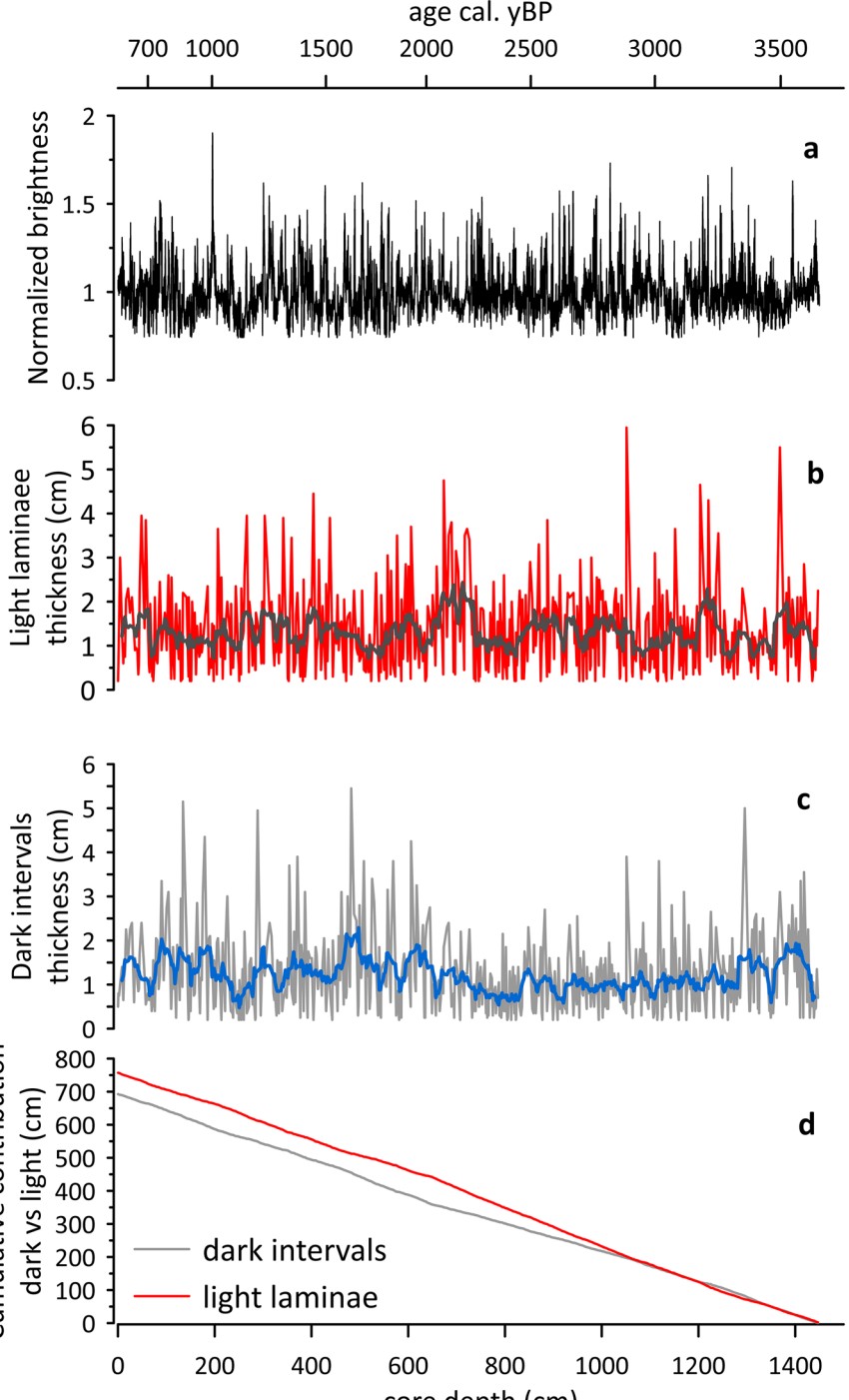

**Fig. 3 | Image analysis outcomes. a** Normalized brightness. **b** Thickness of light laminae. **c** Thickness of dark sediment units. **d** Cumulative depth of light and dark sediments.

08, we applied a five-year averaged greyscale to visually represent the different patterns of fast ice thawing (Fig. 6).

**High-resolution fast ice dynamics and frequency analysis**
Having established that contrasting sediment laminations and colour in Edisto reveal different fast-ice conditions (Figs. 4, 6), we then used grey-scale curves, previously generated at pixel resolution (i.e., at an average temporal resolution of ~0.13 years) (Fig. 3a), to obtain a high-resolution reconstruction of fast-ice dynamics in the Northern Victoria Land with a millennial-scale perspective on fast-ice breakup cycles. In this reconstruction, low brightness values correspond to sediment

deposited during the initial fast-ice breakup while high brightness values indicate protracted and extensive opening of the inlet. To further investigate long and short-term variability, we carried out a frequency analysis on grey-scale data to help identify cycles and fluctuations in fast ice over different time scales (Fig. 7). This allowed us to deconvolve the fast ice trends into components associated with specific frequencies, enabling the detection of long-term trends (low-frequency components) as well as shorter-term variations (high-frequency components). As the age-model of TR17-08 relies on radio-carbon dates, the spectral analysis includes the age uncertainties following the approach proposed by Schulz et al.[34] that was used for

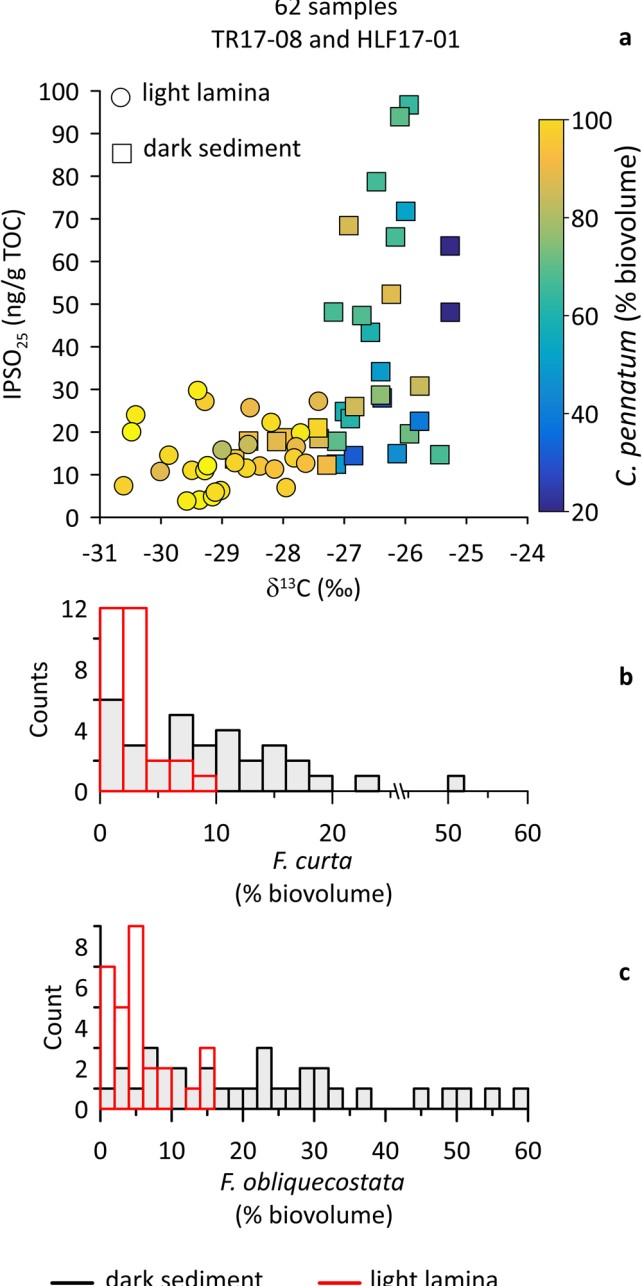

**Fig. 4 | Composition of dark and light sediments subsampled in TR17-08.** Data were merged with HLF17-01 data[26] to obtained 31 samples for each type of sediment (62 samples in total). **a** IPSO$_{25}$, *Corethron pennatum* (biovolume) and δ$^{13}$C. **b** *Fragilariopsis curta* (biovolume, bin size 2%). **c** *Fragilariopsis obliquecostata* (biovolume, bin size 2%).

estimating red-noise spectra from unevenly spaced paleoclimatic time-series. It is also important to emphasize that, with the implementation of this spectral technique, the spectrum was calculated using equally long and overlapping segments of the greyscale data (see Methods) and the average spectrum was then derived from these segments. This approach smoothes the spectrum, particularly in the presence of high-frequency noise. Overall, our analysis revealed two prominent periods at decadal to centennial time scales (Fig. 7a) centred around 90 and 240 years, respectively. To further investigate these low-frequency patterns, we applied a band-pass filter to the grey-scale signals to isolate the 90 y ± 20% and 240 y ± 20% intervals. The band-pass filter analysis confirms the presence of persistent low-frequency cycles of

different intensity where light laminae tend to concentrate (Fig. 7b, c). In other words, periods having an elevated number of light laminae tend to coincide with low-frequency cycles and vice versa. The 90-y cycle is persistent throughout the record, despite exhibiting different intensity, while the 240-y cycle is particularly marked in the upper part of the sediment record.

## Modern fast-ice dynamics and relationship with pack ice

To better understand modern conditions and determine whether the fast ice dynamics observed in Edisto Inlet reflects local dynamics or broader regional climate conditions, we compared the variability of fast ice in Edisto Inlet (Fig. 1b) and pack ice in the northern Victoria Land (Fig. 1a) using MODIS data (Fig. 8; red and blue lines, respectively). The time series indicates that the inlet does not open every year, displaying a distinctive multi-year variability consistent with the laminated pattern. Although the time series is relatively short (i.e., MODIS data have only been available since 2000) on average, the inlet remains closed in about 25% of summers, while in the remaining years, the fast ice thaws either partially or completely. To assess how this temporal pattern ultimately affects sediment deposition, we examined the sediment accumulation rate obtained with $^{210}$Pb in LS23-MUC17-8 core collected at the same location as TR17-08 (Figure S1). LS23-MUC17-8 was retrieved with a multicorer, which better preserve the sediment–water interface. The $^{210}$Pb data show that, despite the multi-year fast-ice variability, sediment accumulation has been continuous on a centennial timescale, consistent with the age–depth model of TR17-08, which displays a fairly linear sediment accumulation rate (Figure S2). This suggests that, in spite of fluctuations in fast ice cover and the presence of laminated sediment structures, there have been no significant interruptions in deposition, such as hiatuses.

Finally, the MODIS-based comparison reveals a strong coupling between the disappearance of pack ice (blue curve) along the Victoria Land coast and the opening of the inlet (red curve) (Fig. 8). When pack ice retreats entirely, especially early in the season, the inlet typically opens fully, for example in 2014, 2017 and 2022. Conversely, when pack ice persists throughout the summer, even in small concentrations, the fast ice in the inlet usually remains intact, such as in 2001, 2008 and 2015. Overall, these findings suggest that fast-ice breakup in Edisto Inlet responds to regional climate driving the pack ice export in the Ross Sea rather than local factors.

## Solar activity as potential driver

Among the triggering forcings that may explain the decadal to centennial-scale cycles identified in the spectral analysis, we considered the potential influence of solar activity, namely Gleissberg (-90-y) and Suess–de Vries (-200-y) solar cycles, which align well with the periodicities observed in the brightness data (Fig. 7). Supporting this, ice core records from McMurdo reveal that the aerosol finger-print, an indicator of atmospheric circulation, is significantly influenced by variations in solar irradiance[35,36]. This points to a pathway by which solar activity could modulate regional climate and, by extension, sea-ice conditions.

A critical piece of this chain is atmospheric circulation, particularly the role of springtime zonal winds. Holland et al.[37] recently demonstrated that variability in zonal winds over the high-latitude South Pacific during October is a strong predictor of the minimum sea-ice extent observed in the western Ross Sea the following March. These zonal winds influence the retreat of sea ice, especially along the Victoria Land coast, where our sedimentary records were collected. The strong coupling between pack ice and fast ice, evident from MODIS data (Fig. 8), provides a mechanistic link between atmospheric drivers and fast-ice conditions in our study area. Notably, these springtime zonal wind patterns are not independent of solar forcing. Mayewski et al.[35], based on NCEP/NCAR reanalyses (1948–2002), found that such wind patterns, particularly over the Pacific sector of the Southern

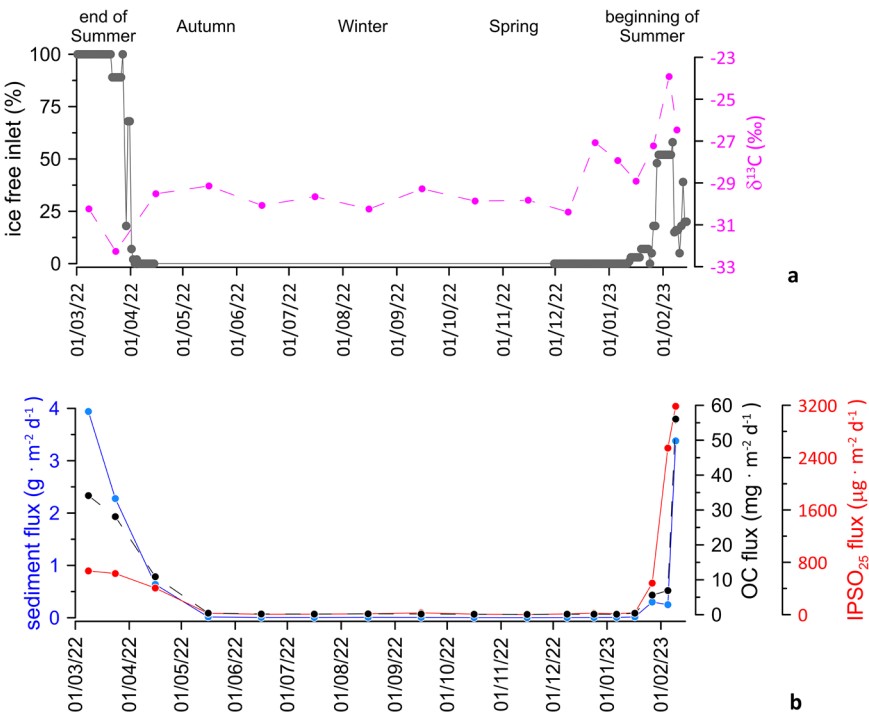

**Fig. 5 | One-year sediment trap data collected in Edisto Inlet. a** Percentage of Edisto inlet ice free (Fig. 1b); stable carbon isotopes ($\delta^{13}C$). **b** Sinking fluxes of sediment, organic carbon (OC) and IPSO$_{25}$ (Ice Proxy for the Southern Ocean with 25 carbon atoms).

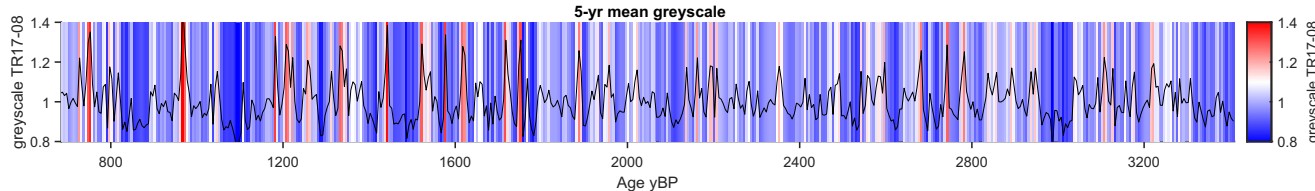

**Fig. 6 | Five-year averaged greyscale record of TR17-08.** Red values indicate prolonged ice-free conditions, whereas blue values represent the initial to moderate opening of the inlet.

Ocean, may themselves be modulated by solar variability. This suggests a cascade of interactions: solar activity influences zonal wind strength[36], which in turn governs sea-ice transport and retreat[37], ultimately affecting the timing and extent of fast-ice breakup along the Victoria Land coast, as recorded in our sedimentary proxy data.

To further explore this hypothesis, we employed high-top chemistry–climate model simulations using CMIP6 solar forcing experiments (CESM2, IPSL-CM6A-LR, CanESM5, MRI-ESM2.0 and HadGEM3-GC31-LL; see Methods)[38] (Fig. 9). While the 4% imposed increase in solar forcing in these simulations exceeds natural total solar irradiance (TSI) variability, these experiments serve as a useful tool for testing the sensitivity of the ocean–atmosphere–cryosphere system to solar forcing. The Solp4p experiment consistently produces a positive Southern Annular Mode (SAM)-like response, with increased mid-latitude pressure and decreased Antarctic pressure (Fig. 9a and Figure S5). This configuration intensifies zonal winds (Fig. 9b), a robust signal across all models, which typically enhances sea ice through northward advection[39]. However, other studies suggested alternative mechanisms wherein intensified zonal winds in austral spring promote sea ice export, especially along the Victoria Land coast leading to increased shortwave radiation absorption, warmer surface temperatures and a reduction in sea ice area[37]. Locally, the model results also show increased northerly winds just equatorward of 70°S in the Ross Sea sector (Fig. 9c and Figure S5), likely associated with a low-pressure

anomaly over the Amundsen Sea. These winds import relatively warm air from lower latitudes, further promoting sea ice loss along the Victoria Land coast. This atmospheric configuration, driven by increased solar irradiance, supports the hypothesis that solar activity can influence fast-ice dynamics through a chain of processes involving sea ice export, atmospheric circulation and ocean warming.

Notably, an increase in the solar constant leads to higher net downwelling shortwave radiation and sea surface warming (Fig. 9d), collectively promoting sea ice melt, making solar radiation the dominant driver in the experiments rather than wind patterns. The resulting sea ice loss reduces the insulating barrier between ocean and atmosphere, enhancing ocean-atmosphere heat exchange and reinforcing warming via a positive feedback loop[40]. Previous studies have shown that this process can lead to increased low-level cloud cover, which traps outgoing longwave radiation and further accelerates sea ice decline[41,42].

## Discussion

Until recently, Antarctic fast ice has been largely overlooked by the scientific community resulting in a limited understanding of its response to climate, effectively becoming one of the "missing pieces of the Antarctic puzzle"[6]. This oversight persists due to evident time and space limitations of observations and simplistic modelling, rendering fast ice a poorly understood component of the cryosphere. Yet,

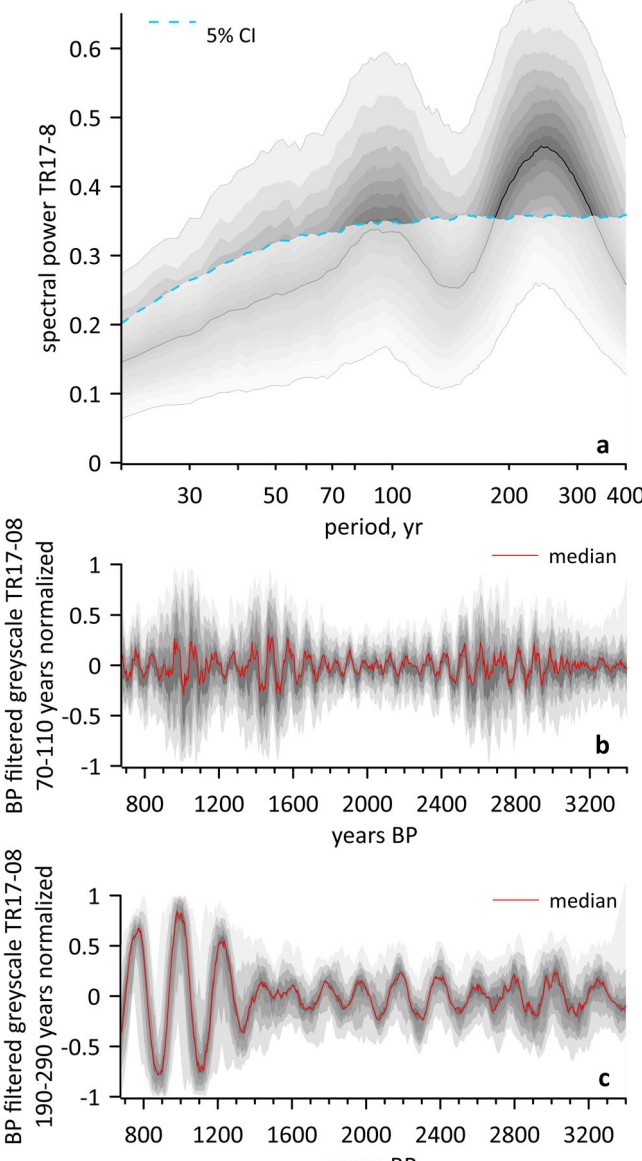

**Fig. 7 | Frequency analysis in core TR17-08. a** Spectral power of the grey-scale record accounting for age uncertainties, based on 1,000 ensemble realizations of our age-depth model; thick black line shows the median spectrum and dashed line indicates the 5th percentile significance threshold across the ensemble. **b**, **c** The continuous line and shadings show the (median) bandpass filtered greyscale record and associated age-model uncertainties, respectively, highlighting the temporal variability of Gleissberg and De Vries solar cycles.

despite its relatively modest coverage, accounting for approximately 4–13% of total Antarctic sea ice[6], its changing coverage and seasonal patterns significantly influence a diverse array of physical, biological, biogeochemical and oceanic processes at local and global scales. In this study, we introduce a novel tool for reconstructing long-term fast-ice dynamics to go beyond the limits of instrumental measurements based on expanded sections of sedimentary sequences collected in fast ice dominated regions. By integrating biomarker and taxonomic analyses with image analyses in laminated sediments at sub-millimetre scale, we achieved an unprecedented high-resolution reconstruction of fast-ice variability in Northern Victoria Land spanning the late-Holocene. Our results clearly indicate that, in addition to multi-year variability (Fig. 7), fast-ice dynamics on the Northern Victoria Land coast exhibit low-frequency patterns possibly linked to Gleissberg and

Suess-deVries cycles, thus pinpointing the potential role of solar forcing in modulating atmosphere-ocean-cryosphere interactions.

A survey of the recent literature indicates that Antarctic fast-ice breakup is significantly influenced by several factors across various spatio-temporal scales[43–48]. Driving factors include snow thickness, interaction with pack ice retreat, air temperature, wind speed and direction, the degree of storminess, basal melt due to the ocean warming, and the incidence of wind-generated ocean waves. For instance, snow in winter promotes the summer breakup of fast ice because increasing snow thickness insulates the ice, reducing its thermodynamic growth rate and consequently its thickness, thus favouring breakup during the following summer[45,49]. On the other hand, snow also contributes to sea-ice thickening by triggering snow-ice formation when heavy snow causes surface flooding[45,49]. In-situ fast ice melt due to atmospheric temperature appears to be a minor factor across Antarctica, while the mechanical breakup, driven by a combination of wind, waves, and warming of ice-free adjacent waters, seems to be a primary factor, especially when pack ice no longer provides protection to the fast ice[6,10,43,47,50,51]. Another type of atmospheric forcing driving fast-ice breakup is the occurrence of local katabatic winds that push fast ice off the coast in summer, whereas in winter they contribute to the regulation of snow cover and, consequently, the thickness of fast ice[6,45]. An additional, poorly investigated driver is the intrusion of Modified Circumpolar Deep Water (mCDW) inside the fjords. Indeed, the inflow of mCDW onto the Antarctic continental shelf is a key driver of Antarctic ice shelf mass loss and, thus, it could potentially affect coastal fast-ice breakup[52,53]. To the best of our knowledge, there is no documented evidence of modified Circumpolar Deep Water (mCDW) intrusion into any of the fjords along the Northern Victoria Land coast. Yet, the intrusion of mCDW has been inferred for the Edisto Inlet, based on the presence of the benthic foraminifera *Epistominella exigua* from 2 ka to 1.5 ka[54].

Finally, retreat patterns of the pack-ice edge are found to be key factors that favour fast-ice breakup. Pack ice acts as a barrier reducing exposure to solar radiation, surface winds and ocean waves. In our study region, MODIS time series data clearly show that when even a small percentage of pack ice persists (about 10%) along the Victoria Land coast during summer, fast ice in Edisto Inlet remains intact (Fig. 8). In contrast, when pack ice is efficiently exported and melts early in the season, Edisto Inlet experiences a prolonged opening, resulting in the deposition of light laminae rich in *C. pennatum* and marked by distinct geochemical signatures (Fig. 5). This clearly indicates that the Victoria Land coastal region is highly sensitive to the timing and extent of pack-ice minima during summer. A recent study[37] highlights that variability in spring zonal winds over the South Pacific strongly predicts sea-ice extent minima in the western Ross Sea the following March. These winds influence ice movement and retreat near Victoria Land, affecting fast ice stability in the region. Another study[35] showed how the TSI may itself influence the zonal wind patterns. Together, the current literature suggests a chain of influence wherein solar variability affects zonal winds, which then impact sea-ice behaviour and ultimately fast-ice breakup along the Victoria Land coast.

The role of zonal winds in relation to TSI variability is further supported by transient climate simulations over the last millennium, which show a strong decline in the Southern Annular Mode (SAM) index and weak zonal winds during the Spörer (1388–1558 CE) and Maunder (1621–1718 CE) Grand Solar Minima[55]. Paleoclimate evidence from core HLF17-01 (Fig. 1) aligns with this signal, showing limited inlet opening during these periods[26], as indicated by a distinct geochemical fingerprint based on the same diagnostic tools used in this study (Fig. 4a). Although core TR17-08 lacks its uppermost section, the HLF17-01 record[26] (Fig. 1) provides supporting evidence that periods of reduced solar activity, such as during this minimum, may exert cascading influences on atmospheric circulation, sea-ice dynamics and the stability of coastal fast ice in our study region.

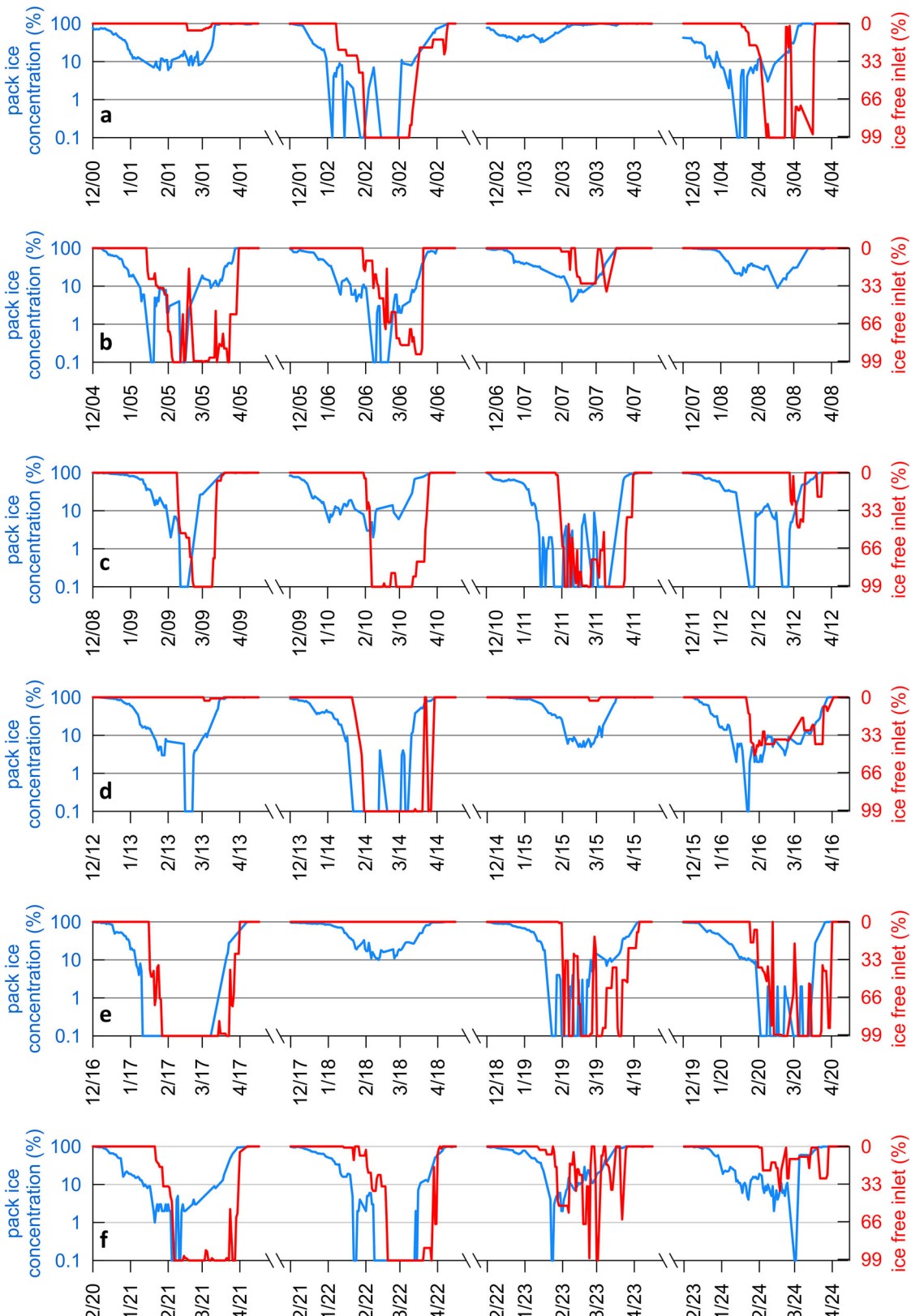

**Fig. 8 | 24 y long times series of pack-ice (blue line) and ice-free inlet concentrations, see Fig. 1. a** Dec 2000–Apr 2004; (**b**) Dec 2004-Apr2008; (**c**) Dec 2008–Apr2012; (**d**) Dec 2012–Apr2016; (**e**) Dec 2016–Apr2020; (**f**) Dec 2020-Apr2024.

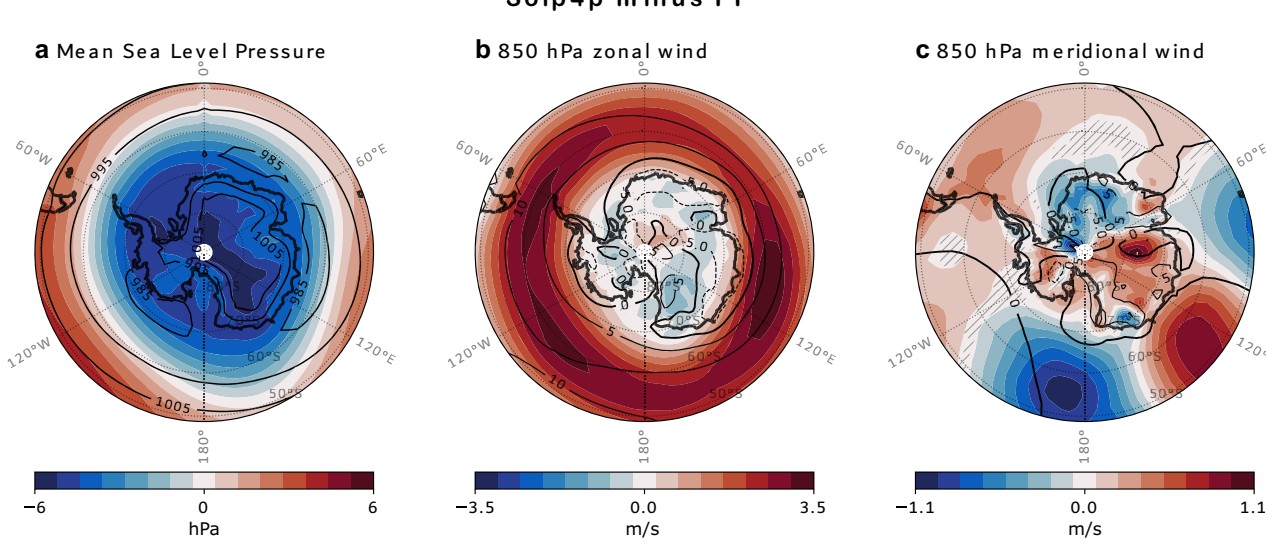

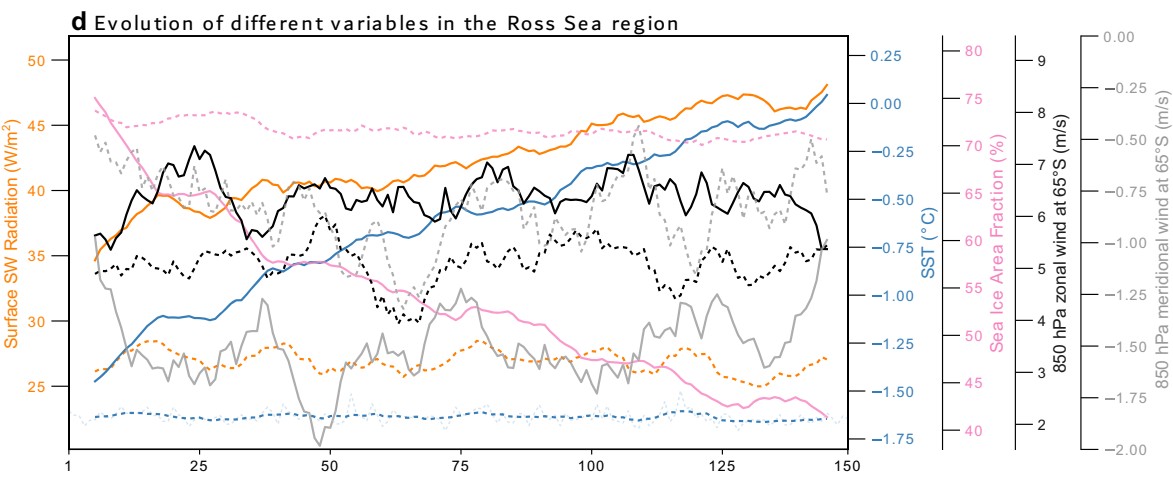

**Fig. 9 | Multi-model mean, annual mean anomalies (Solp4p minus PI). a** Sea level pressure, (**b**) zonal and (**c**) meridional winds at 850 hPa. Black lines in (**a**–**c**) show annual mean results in the PI experiment, while hatching indicates anomalies that are not statistically significant at the 95% confidence level, based on a two-sided t-test. **d** Yearly evolution of the net surface shortwave radiation, sea surface temperature and sea ice area fraction, averaged over the Ross Sea, defined as the region between 160°E–130°W and 70°S–85°S. 850 hPa zonal and meridional winds at 65°S, averaged over the same longitude range, are shown in green and grey lines. Solid and dotted lines in (**d**) show 10-year running mean from the Solp4p and PI experiments, respectively.

Further studies of past climate have highlighted the role of TSI in modulating the zonal winds at low southern latitudes. For example, Varma et al.[56] based on proxy data and climate simulations, suggested that solar variability during the Holocene influenced the position of the southern Hemisphere Westerly Winds (SWW), with lower solar activity linked to equatorward shifts and higher activity to southward shifts. Riechelson et al.[57] have shown that precipitation variations in Patagonia during the late-Holocene, influenced by the latitudinal shifts of the SWW, display significant centennial cycles that align with Gleisberg and Suess-de Vries solar cycles. The authors also propose a mechanism whereby southward (northward) SWW movement in response to increasing (decreasing) total solar irradiance cools (warms) Antarctic air temperatures due to upwelling, which however should increase the CDW advection over the Ross Sea margin. Changes driven by solar activity are not limited to atmospheric forcing but can propagate in the ocean down to intermediate water masses. For instance, Seidenglanz et al.[58] used a comprehensive global climate model to study the impact of Gleissberg and Suess-de Vries cycles and found significant

sensitivity of ocean temperatures, particularly in the southern Atlantic sector, down to intermediate waters, driven by a latitudinal shift of the westerly wind belt.

Our analysis of CMIP6 experiments provides further perspectives on the atmospheric processes influenced by solar forcing that can contribute to the regulation of fast-ice variability (Fig. 9). It is important to note that the solar forcing applied in these simulations (4%) significantly exceeds natural variability; therefore, these results should be interpreted as offering mechanistic insights rather than direct evidence, complementing findings from re-analyses[35,37]. Our results further support the substantial modifications in zonal and meridional wind patterns, consistent with outcomes from reanalyses[37] and paleoclimate reconstructions[35] (Fig. 9b, c). However, in the Solp4p experiments, other factors, such as increased solar radiation and the resulting sea surface warming, emerge as main drivers of fast-ice breakup (Fig. 9d). The influence of solar forcing in the Ross Sea was examined by Stewart et al.[59] in the context of sea ice dynamics and ice shelf stability. They showed that it plays a key role in Antarctic sea ice dynamics by driving polynya formation and associated heat exchange.

Seasonal sea-ice retreat exposes open-water areas like the Ross Sea Polynya to increased solar radiation, leading to substantial surface warming that delays sea ice growth and forms a warm, buoyant layer capable of impacting nearby ice shelves.

To conclude, our novel approach based on multi-proxy analysis of laminated diatomaceous sediments, provides a viable means to extend instrumental records of Antarctic fast-ice dynamics further back in time than has been hitherto unavailable. We have demonstrated that the frequency of fast-ice breakup does not follow a simple yearly cycle but exhibits a more complex pattern on longer timescales possibly modulated by solar forcing and perturbation of zonal winds with cascade impacts on the ocean-cryosphere-atmosphere interactions. We envision a chain of events in which sea ice export driven by zonal wind exposes the fast ice to local wind, waves, and thermodynamic processes. Given that laminated sediments are a common characteristic of Antarctic marine archives[24], our innovative approach holds potential for broader application in other Antarctic regions and over extended time frames in the future. Additional research using a similar approach in laminated and expanded deposits in other fast ice-dominated regions is thus needed to further investigate the role of natural forcing on fast-ice dynamics and its influence on a pan-Antarctic scale.

## Methods

### Sediment core sampling and handling

Cores TR17-08 was collected on board of the *M/N Italica* in 2017 using a piston corer at a depth of 462 m (72° 18.2778′ S, 170° 04.1784′ E). The coring site was revisited in 2023 with a multi-corer (LS23-MUC17-8) on board of *I/B Laura Bassi* using the multi-corer, at a depth of 462 m. Following retrieval, both sediment cores were stored at 4 °C during transport and prior to laboratory processing. Subsampling was carried out in the lab, after which samples were freeze-dried for subsequent analyses.

### Age constraints

The age-depth model of LS23-MUC17-8 was established using excess $^{210}$Pb. The activity of the short-lived radionuclide $^{210}$Pb was assessed indirectly by measuring its decay product, $^{210}$Po, through alpha spectrometry. Approximately 1 gram of sediment was used for the extraction of $^{210}$Po. The samples were initially dried, homogenized, and weighed, followed by acid digestion using a concentrated mixture of $HNO_3$ and HCl to dissolve the sediment matrix. A $^{209}$Po tracer was introduced as an internal standard to evaluate the recovery efficiency. Post-digestion, the solution was evaporated and $^{210}$Po was isolated via spontaneous deposition onto silver discs. Sediment accumulation rates (SAR) were then calculated using the constant flux–constant sedimentation model[60].

The age model of core TR17-08 was constructed using 10 radiocarbon determinations derived from carbonate tests and the Mount Rittmann tephra layer (Table S1). For the tephra layer, we used the calendar age assignment derived from the annually-counted section of the WAIS Divide 2014 chronology[31]. The absence of excess $^{210}$Pb suggests that the upper section of TR17-08 sediment record is missing. Consequently, the age–depth model was conservatively constrained between the tephra and the lowest radiocarbon date as conservative approach. To quantify age uncertainties, the age constraints were used as input to a Bayesian age-depth model using Bacon2.3 (ref. 61). Prior to age modelling, the $^{14}$C determinations were calibrated against the Marine20 calibration curve[62] and corrected for a local reservoir effect (ΔR) of $663 \pm 35$ y. The ΔR was estimated leveraging on the calendar age of Mount Rittmann tephra and the radiocarbon date located 6.5 cm away from the tephra horizon. The age gap was adjusted using the sedimentation rate of LS23-MUC17-8. The estimated ΔR fits within uncertainties of the ΔR estimated for the Ross Sea[63] with, however, a lower absolute error.

The Bayesian age-depth model was run using default parameters except for the accumulation rates priors, i.e. *acc.mean* and *acc.shape*, which were set to 2 and 0.5, respectively, to allow the model to effectively simulate the sharp change in sedimentation rates around 55 cm and a *size* parameter of 8k to allow for a sufficiently long MCMC simulation. The final age model (Figure S2) produced a stable Markov chain Monte Carlo (MCMC) run and is qualitatively comparable to that presented in Tesi et al.[26] for its twin core HLF17-1.

### Organic geochemistry

Total organic carbon (TOC) and stable carbon isotopes (δ$^{13}$C) were analysed using a Fisson2000 Elemental Analyzer coupled with a DeltaQ Mass Spectrometer via COFLO IV (Thermo-Fisher). Before TOC and δ$^{13}$C samples were acidified with 1.5 M HCl in silver capsules to remove the inorganic carbon. Reference $CO_2$ was calibrated using replicate measurements of IAEA-CH7 external standard. Uncertainty (1-sigma) was lower than 0.1‰ based on replicate analyses of in-house standards. TOC was assessed based on acetanilide external standard and the coefficient of variation (1-sigma/mean) was lower than 5% based on replicates of internal standard.

The analytical procedure for the Antarctic sea ice biomarker proxy IPSO$_{25}$ assessment remained consistent with previous methodologies[26,33]. Before the extraction, 9-octylheptadec-8-ene (9-OHD) was added to sediments as an internal standard for subsequent quantification of IPSO$_{25}$ via gas chromatography–mass spectrometry (GC-MS). Sediments underwent saponification in methanolic KOH ($H_2O$/MeOH, 1:9; 5% m/v KOH) at 70 °C for 60 min. Following saponification, organic biomarkers were extracted with hexane (3 × 3 ml) and the resulting supernatant was dried under a $N_2$ stream. The dried extract was reconstituted in hexane (500 μl) and purified via open column chromatography ($SiO_2$, 38–63 μm). The eluted hexane fraction (3 × 2 ml), containing highly branched isoprenoids (HBIs; including IPSO$_{25}$), was subsequently dried under a $N_2$ stream and reconstituted in 300 μl of hexane prior to GC-MS analysis. IPSO$_{25}$ quantification was performed using an Agilent 7820a chromatograph equipped with a J&W DB5-MS column (30 m length, 0.25 mm i.d., 0.25 μm film thickness) coupled to a 5977b Mass Selective Detector (MSD). The oven temperature ramp ranged from 60 °C to 300 °C at a rate of 5 °C/min. Throughout the ramp, the MSD operated in both Selective Ion Monitoring (SIM) and SCAN modes.

Identification of IPSO$_{25}$ relied on comparison of its mass spectrum and GC retention index with previously published data[64,65]. Injection of C8-C40 alkanes (Sigma-Aldrich) was utilized to confirm the retention index of IPSO$_{25}$ (RI 2082[64]). Quantification was achieved by integrating peaks of ion m/z 348.3 in SIM mode, normalized to the corresponding peak area of the internal standard (9-OHD) and adjusted using an instrumental response factor obtained from analysis of a purified in-house standard.

### Diatom assemblages

Diatom analysis commenced with treatment of approximately 0.2 g of dry sediment in a beaker containing distilled $H_2O$ (40 ml per sample), $H_2O_2$ (60 ml per sample; concentration 40%) and $Na_4P_2O_7$ (100 mg) to dissolve organic matter and disaggregate sediment particles. The suspensions were heated at 70 °C for 45 minutes, followed by addition of 10 ml of HCl (concentration 10%) to remove carbonates. Further heating at 70 °C for 15 min ensued and subsequent rinsing with distilled water was performed until a pH of approximately 5–6 was attained. Rinsing occurred every 8 h to facilitate diatom settling. The resulting suspensions were then reduced to a volume of 50 ml.

For microscopic examination, a coverslip was positioned inside a Petri dish and a measured volume of suspended material (approximately 150–300 μl) was pipetted onto it, along with distilled water to ensure a heterogeneous distribution of diatoms. After excess water removal, coverslips were affixed to microscope slides using Norland

Optical Adhesive 61 (NOA61) and dried under UV light. Diatom counting followed a methodology proposed by Crosta and Koç[66] with at least 300 diatom frustules counted for each slide. In calculating the relative abundance of each diatom species, valves of *C. pennatum* were counted as one when more than half of the valve was present. This counting approach was based on the method suggested for other genera by Crosta and Koç[66]. The absolute diatom abundance was not considered in this study due to its susceptibility to non-uniform diatom distribution on cover slips. Additionally, relative biovolume contribution was calculated based on individual biovolumes proposed for each species by Alley et al.[24].

## Image analysis

To quantify the laminations displayed in Fig. 2 and Figure S3, we used two software packages described in Weber et al.[28]. The software consists of a set of Visual Basic macros that are executed from within MS Excel. We used version 2 available at https://doi.pangaea.de/10.1594/PANGAEA.775955. First, we took the original line-scan (surface) images of every 1-m section from cores TR17-08 and reduced the original resolution of each tiff image from ca. 15,000 pixel to ca. 2000 pixel in length to make the analysis more manageable while still maintaining sufficient resolution. Then, we converted those images to the required lossless bmp format for the BMPIX tool and defined a centre line along the core to generate a grey-scale curve. This was achieved by averaging 30 pixels perpendicular to the line for each centre pixel to reduce noise, resulting in a grey-scale curve consisting of ~2000 downcore data points for each 1 m section. Accordingly, we collected 29,001 grey values for core TR17-08 (Fig. 3a).

In the next step we used the PEAK tool to count the laminations automatically based on three different methods (Figure S3). First, the maximum count algorithm counts every bright peak of a couplet of two laminae in a smoothed curve following the definitions that have to be made by the user – the full width half maximum (FWHM) of the Gaussian smoothing, the minimum cycle length and the minimum amplitude. Second, the zero-crossing algorithm provides separate counts for every positive (bright) and negative (dark) halfway-passage of the curve through a wide moving average, thereby providing single laminae resolution. The same is true for the frequency truncation method, which relies on Fourier transformation to decompose the curve into its frequency components before counting positive and negative passages (Fig. 3b, c). An example of the individual steps and procedures used by the BMPIX/PEAK software package and the results are shown for segment VI of Core TR17-08 in Supplementary Fig. S3.

Compared to rather subjective manual counts, the software offers the advantage of being highly objective, reproducible and relying on mathematical criteria. Since all information is displayed graphically, interactive optimization of the settings can be achieved quickly and conveniently. Also, by using a total of up to five different count algorithms we are able to evaluate uncertainties and to study the thickness variability of bright and dark laminae separately.

## Spectral analysis

To identify individual band cycles in our sedimentary records, we employed spectral analysis using the function redfit[34] of the dplR package in R[67]. We ran the analysis on the portion of the core between the upper tephra horizon and the last radiocarbon date, as a conservative approach where the age constraint is more robust. The function estimates the red-noise corrected spectrum of an unevenly sampled proxy timeseries using the Lomb-Scargle Fourier transform. The significance of spectral peaks is tested against the red-noise background from a theoretical AR-1 process. The function splits the timeseries into n segments, which overlap by 50%, whereby the final spectrum is the average of the n periodograms. For our analysis, we used a value of $n = 4$. To propagate age uncertainties into the spectral analysis, we estimated the spectrum of the grey-scale data and its

significance, from 1000 independent realizations of the age model (Fig. 7a), following the approach outlined above. To determine the minimum significance threshold across the 1000 spectra, we used the 5th percentile, providing a conservative yet reasonable criterion for assessing whether a spectral feature is robust. To account for the uncertainties with our age-model in the bandpass filtering, out of the 1000 age model ensembles, we extracted the subset in which the solar spectral (deVries and Suess cycles) peaks exceeded the significance threshold, and applied the bandpass filter to the greyscale record on each one of those chronologies (7b, c).

## Sea ice data analysis

To investigate temporal variability of fast ice in the Edisto Inlet and pack ice in the coastal region adjacent to the inlet (polygons, Fig. 1), we applied an automated analysis procedure based on thermal infrared data from the MODIS sensors on board NASA's polar-orbiting Earth Observing System (EOS) satellites, i.e., Aqua (operational since 2002) and Terra (since 2000). Analysis of satellite imagery consistently shows that sea ice within Edisto Inlet is classified as landfast ice (Supplementary Material, Figure S4). In detail, we processed Level-1B granules obtained from NASA's MODIS Atmosphere Archive and Distribution System (LAADS). These swath-based scenes, acquired multiple times per day, offer a spatial resolution of 1 km × 1 km at nadir and cover approximately 1354 km across-track and 2030 km along-track.

To retrieve ice surface temperature (IST) and reliably distinguish between sea ice and open water, we used several MODIS products: MYD/MOD02 (radiance), MYD/MOD03 (geolocation) and MYD/MOD35 (cloud mask). The latter also provides atmospheric information essential for identifying scenes affected by cloud and fog. Our analysis focused on austral summer observations (1 December–15 April) from December 2000 to April 2024. We extracted all relevant scenes covering Edisto Inlet and its adjacent region (Fig. 1) and applied land–ocean discrimination using coastline data from NOAA's National Geophysical Data Center for the Ross Sea.

For each MYD02 scene, corresponding MYD35 data were examined to identify and discard scenes compromised by atmospheric interference. The cloud mask assigns each pixel a confidence level among "confident clear" (>0.99), "probably clear" (>0.95), "uncertain" (>0.66), and "not clear" (≤0.66). We calculated the proportion of oceanic pixels within each scene falling into each category. Scenes in which more than 80% of ocean pixels had a confidence level above 0.66 were retained for further analysis; all others were excluded. IST was then calculated for each valid oceanic pixel and used to classify sea surface conditions as either open water or ice-covered, based on temperature thresholds[68]. In clear-sky conditions, open water is easily identifiable due to its warm and homogeneous temperature, typically above the saline water freezing point. When scenes are fog-contaminated, IST estimation is still possible but more challenging[69]. Nevertheless, the presence of strong gradients of temperature associated to the transition between open water and ice pack allows for reliable ice/water discrimination. Additionally, suspect scenes, in which low consistency is found with preceding and following acquisitions, were also visually inspected and discarded when not reliable. This procedure allowed us to obtain near-daily observations of the Edisto Inlet and the surrounding area throughout most of the study period, enabling sea ice retrieval for approximately [96%] and [79%] of the total time span, respectively.

## Sediment trap

An oceanographic mooring was deployed in Edisto Inlet between March 2022 and February 2023 at 468 m water depth (72° 18.520′ S and 170° 03.187′ E). The mooring was fitted with a sediment trap at 224 m from the sea surface. Seventeen samples of sinking particles were collected and analysed for TOC, $\delta^{13}C$ and IPSO$_{25}$ analyses as presented in Section 4.2.

## AbruptSol experiments

To understand the response of the Southern Hemisphere surface ocean and atmospheric circulation to a change in the solar constant, we analysed model outputs from the Cloud Feedback Model Intercomparison Project (CFMIP)[70], which is part of the sixth phase of the Coupled Model Intercomparison Project (CMIP6)[71]. We analysed the CFMIP 'abrupt-Solp4p' experiments (Solp4p), where fully coupled climate models initialised from pre-industrial (PI) conditions were perturbed with an abrupt 4% increase in the solar constant while keeping the distribution of solar energy across different wavelengths consistent with the PI experiment. These experiments were run for 150 years under this constant solar forcing, with other forcings, such as greenhouse gas concentrations, held at PI levels. Therefore, comparing the solp4p experiments with the PI control experiments allows us to examine changes in sea surface temperatures and sea ice cover around Antarctica brought about by increase in solar radiation.

We analysed the monthly mean atmosphere and ocean outputs from the Solp4p and PI experiments conducted using: the Community Earth System Model version 2 (CESM2)[72], the Institute Pierre-Simon Laplace Climate Model version for CMIP6 in the low-resolution configuration (IPSL-CM6A-LR)[73], the Canadian Earth System Model version 5 (CanESM5)[74], the Meteorological Research Institute Earth System Model version 2.0 (MRI-ESM2.0)[75] and the low-resolution configuration of Hadley Centre Global Environment Model version 3.1 (HadGEM3-GC31-LL)[76]. Table S2 specifies the horizontal resolutions of the atmosphere and ocean components of the models used in this study. For multi-model mean calculations, the atmosphere and ocean model outputs were re-gridded to commonly used 1° × 1° resolutions using bilinear interpolation.

## Data availability

All pertinent data supporting this study's findings are freely available as Supplementary Information.

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

## Acknowledgements

We thank the captains and the crew of *N/R Italica*, *N/R OGS Explora* and *I/B Laura Bassi* for their support during the sediment coring and geo-physical surveys. This study was supported by the following PNRA projects: TRACERS (PNRA16_00055-TefRchronology and mArker events for Correlation of natural archives in the RosS Sea, Antarctica) principal investigator A.D.R., EDISTHO (PNRA18_00010-Edisto Inlet DIatom lami-nations Sequences Through the Holocene) principal investigator K.G. and LASAGNE (PNRA19_00069-Laminated sediments in the magnificent Edisto Inlet (Victoria Land): What processes control their deposition and preservation?) principal investigator LL. M.E.W. received funding for this research by the Deutsche Forschungsgemeinschaft (DFG-Priority Pro-gramme 527, Grants We2039/17-1 and We2039/19-1). FM is supported by a NERC Discovery Science Grant (NE/W006243/1).

## Author contributions

T.T., F.M. and M.E.W. designed the study. M.E.W. performed image analyses of core TR17-08. F.M. developed the age-depth model of core TR17-08 and performed frequency spectrum analysis. D.D. analysed outcomes from climate models. L.S. and K.G. quantified diatom assemblages. C.P. and T.T. identified and quantified the IPSO$_{25}$ bio-marker. F.B. measured stable carbon isotopes. A.G., E.C., L.L. and A.D.R. collected core TR17-08. P.G., E.C., C.M., K.G., A.D.R., and A.G. oversaw the processing and curation of the core samples. G.A. performed MODIS analysis. T.T. wrote the manuscript, with contributions of A.D.R., A.G., C.M., C.P., D.D., E.C., E.C., F.M., G.A., K.G., K.G., L.L., L.S., M.E.W., P.G.

## Competing interests

The authors declare no competing interests.

## Additional information

**Peer review information** *Nature Communications* thanks Xavier Crosta, Alexander Fraser, and the other, anonymous, reviewer(s) for their con-tribution to the peer review of this work. A peer review file is available.

