## [Transparent Peer Review file · Nature Communications]

Late Holocene fast-ice dynamics around the Northern Victoria Land coast, Antarctica

Corresponding Author: Dr Tommaso Tesi

Version 0:

Reviewer comments:

Reviewer #1

(Remarks to the Author)

Together with an ECR, as encouraged by NComms, we have now finished a review of NCOMMS-24-83194. As required, we together submit this single report incorporating both of our views on the manuscript.

This manuscript represents an incredible amount of rigorous work - including a marine science voyage, sediment core extraction, sediment dating, rigorous work defining layers within the sediment, diatom analysis to determine overlying ice conditions, spectral analysis of the record, and comparison with CMIP6 data simulating a different solar constant. While the details of the sediment work are a little out of our area of expertise, we found no major problem with what was presented in that regard. We did, however, find some major shortcomings with the methodology which aims to provide evidence of the insolation cycles being the primary driver of fast-ice stability, and these are outlined below.

Major Comments

There are time series presented of band-passed insolation and sediment brightness (Fig 5 b and c), and it is argued that insolation modulates the sediment greyscale (and thus fast ice). However, no correlation is performed between these. Without such an obvious correlation being performed, the reviewer gets sceptical that the correlation is low and/or insignificant. If that were to be the case, then the entire causality argument hangs on an argument that the periodicity is similar, so it must be the driver. This is not really strong enough evidence to make this claim. The periodicity isn't even a match for the middle period discussed (90 vs 110 years), and for that band the 110 year period isn't even very prominent. I may be wrong, and the correlation might prove significant, but eyeballing Fig 5c, it doesn't look promising. Fig 5b maybe but it's too hard to eyeball it (plus there's a spectrum mismatch for this band, as noted above). Basically it's not a strong argument without a correlation. And the argument is the crux of the manuscript (it's in the title). Similarly, it's stated a couple of times that the shorter periodicity is due to IPO or AMO variability. There's even less evidence presented for this (no analysis at all, just a hypothesis), but it's upheld as fact in places (e.g., lines 169, 299). This is very much overstated and should be heavily caveated.

L264-277: The discussion on mechanism here is lacking in evidence. Katabatic wind representation in CMIP class models is poor and unreliable. There's also no evidence given to show that downwelling shortwave is the primary factor for the reduced pack-ice extent. Regarding the comment on snow cover, Ushio (2006) showed that a deep snow cover favours stability, i.e., in the opposite sense to the hypothesis given here. This lack of evidence is summed up in the first sentence of the next paragraph (L279). The CMIP simulations do provide evidence that the surrounding environment changes with +4% insolation, but the links to fast ice are too nebulous, without any correlation presented between insolation and the sediment colour.

Figure quality is questionable - all except fig 4 display compression artefacts. Fig 1 is particularly poor. Inset c has no scale and includes illegible text. The "satellite image" has no more information on which satellite. It shows fast ice covering the region in January, which is a bit misleading (this region has fast ice breaking out most summers). Panel d has illegible axes, and somehow two x-axis labels which aren't explained. The right scale has no label. The caption confuses the orange and green dots (I think they should be switched in the caption). "This study" is ambiguous in the caption. The "red line" is almost not red due to compression. Fig 2 and 3 are fine aside from the compression of text. Fig 4 has a poor choice of colour bar (is yellow 20 or 100?) making interpretation of that very difficult. Fig 5, as mentioned above, suffers from poorly-labelled axes, often missing units. The "highpass" is interesting and completely ignored in the method - why was a high pass needed? In panel a why was the 30 year period never annotated? Fig 6a is very confusing. What is "year"? Year since the start of the

AbruptSol experiment? It's very unclear. The orange label is clearly wrong - I guess "increase vs control" is needed (insolation should be over 100 W/m² even averaging over a cloudy year)? Is this averaged over a year? PI is not defined in the comment. There seems to be interchangeable use of "solp4p" and "AbruptSol" - can't these be combined? Panel c's colour bar doesn't give the "sense" (e.g., red is northward) but I concede it's possible to figure this out by carefully reading the caption - but why not make it easier for the reader? I paid less attention to the supp figures, but S2 is really illegible when printed. The purples get completely lost and the blue isn't sharp at all. Some of these aspects are described in greater detail below in the "minor comments" section.

Minor Comments

Line 1 - coast is a descriptor of the location not part of the name thus should not be capitalized

Line 27 - "Fast-ice" should not be hyphenated in this scenario. In this scenario "fast ice" is a noun and should not have a hyphen. "Fast-ice" should be used as an adjective ex. fast-ice extent. This problem persists throughout the document

Line 32 - Antarctica is misspelled

Line 32 - The use of the oxford comma is inconsistent throughout the document

Line 32 - "image analysis at high resolution" does not match the description later in the document of a high resolution image analysis.

Line 34 - solar activity not defined (sunspots?), and CMIP6 not properly defined.

Line 55 - "mixed" is normally "modified" and "waters" is normally "water" here.

Line 58 - fast ice has been shown to modulate glacier tongues, not tidewater glaciers.

Line 59 - not seal breeding - just foraging (as far as I know).

Line 70 - I don't agree with "highly approximated" here - the fast ice simulations are quite realistic, but the downside is that they are only regional now.

Line 71 - referencing the wrong Fraser paper (should be 2020 for the dataset)

Line 80 - show-case should not be hyphenated used showcase

Line 83 - "consolidated" isn't enough of a description. Actually, this fast ice breaks out most years in recent times, and this isn't really stated clearly anywhere.

Line 97 - The introduction does not include descriptions of the Gleissberg and Suess-de Vries solar cycles. The introduction would benefit from information about these solar cycles given a lot of the following information and conclusions rely on an understanding of what these cycles mean and their time scales.

Line 108 -109 - Missing error value " $3.7 \pm ka BP$ ". There should be a value after the \pm to establish the time error.

Line 133 -155 - The content is synthesis of the results and better fits in a discussion section

Line 150 - "more or less" is ambiguous.

Line 157- "fast sea ice" is new terminology not used elsewhere.

Line 158 - Edisto needs to be Edisto Inlet to improve clarity and allow the reader to reference themselves within Figure 4

Line 165 - should be "deconvolve"

Line 198 - "Overall, although" is awkward phrasing

Line 200 - 204 - Figure 6a, which is what this text is describing, displays increasing surface shortwave radiation, SST, and precipitation which coincide with a decrease in sea ice area fraction. There are connections made between the variables which are logical but no evidence that they have any dependence or affect on each other is provided. This can be done in the form of citations which found the connection or statistics done on this data.

Line 211 - 212 - this reference indicates sea ice extent increases by a positive SAM, not a concentration

Line 216 - "wind-driven sea ice loss is negligible" is not shown anywhere.

Line 225 - This statistic is from Fraser et al. 2021 and not cited

Line 226 - double use of "and" is not grammatically correct

Line 236 - 260 - Paragraphs are introduction material not results.

Line 236 - I was expecting a citation to support the statement following made in the sentence which starts "A survey of the recent literature indicate"

Line 264 - there appears to be a 'that is a typo at the beginning of this line

Line 264 - 267 - This sentence is confusing and needs to be revised. Suggestion to make break into shorter sentences

Line 269 - 273 - no evidence is provided to substantiate the conclusion that increase in downwelling shortwave radiation is the primary driver of pack ice loss. SST, and precipitation also increase and no evidence was provided to quantify the importance of each driver

Line 277 - No evidence is provided to substantiate the claim the precipitation of snow is a first order effect in the context of this paper.

Line 282-283 - the use of collectively and operating together mean the same thing and do not need to both be said

Line 283 - When a statement like "a survey of current literature" is made I would expect to see a citation to substantiate the second half of the sentence.

Line 291 - need to indicate if this effect applies to Antarctic air or sea surface temperatures

Line 294 - Southern should not be capitalized as it described a location and is not the name of the location

Line 313 - ΔR needs to be in either (ΔR) or commas as it is an abbreviation of reservoir effect and used later as a stand alone representation of reservoir effect.

Line 319 - MCMC simulation is an abbreviation and not defined prior to being used

Line 334 - need space between numerical value and the units

Line 382 - Comma misplaced. Should be after "manual counts" instead of "rather subjective"

Line 406 - solp4p should be capitalized since it is capitalized throughout the text

Figure 1 - The orientation of 1a and 1b are inverted and there are not visual indications of this

Figure 1c - Need some form of scale

Figure 1d - Tick labels all very small and illegible

Figure 1d - left and right axis are the same, don't need both

Figure 4a - Color bar wraps where values of 20 and 100 have similar colors. This is misleading and the color scale need to not wrap when the data values do not wrap

Figure 4 b and c - the x-axis values are misleading when comparing these plots. A bar with the same width represents 2 % bin of biovolume in b but 3.33% bin in c. Are the bins different sizes?

Figure 5 - Axis labels could be simplified

Line 1 - coast is a descriptor of the location not part of the name thus should not be capitalized

Table S1 - How are you resolving a 30 year cycle for IPO when the error is between 23-91 years?

Figure S1 - Both x and y axis labels and tick labels need to be larger

Figure S1 - x axis needs more labels when, more frequent than ever 500 cm

Figure S1 - What are the blue spikes?

Figure S2 - Purple lines are very hard to see with the dark image as the background

Figure S2 - the floating average displayed in S2a is significantly smoother than the floating average in b and c. Please confirm the floating average in each figure is calculated the same.

Line 534: Lemieux, J. should be Lemieux, J.-F.

Reviewer #2

(Remarks to the Author)

The data sets presented in this paper are interesting and can provide insight into multidecadal influences on ice dynamics and possibly an opportunity to learn about regional climate connections. Deeper consideration into the mechanism around the observed variability could expand the paper's impact/relevance by connecting it to the broader setting as well as focus the paper's narrative. Additionally, for frequency analysis, decisions regarding the age model need to be better supported.

Major Comments:

More about the modern setting could help support the paleo interpretation/mechanism.

Something similar to Figure 5 in Tesi et al., 2020 could be useful in demonstrating: 1. Seasonal progression of sea ice melting and 2. Interannual variability in inlet opening. In L76-77, it is mentioned that Fraser et al has a few decades of satellite data. Could the authors compare inlet opening in the satellite images with the TSI and/or SAM index to see the correlation (does the inlet open in years when TSI increases or decreases? Positive or negative SAM?).

Is there any sediment trap data at all from the broad area? It would be helpful to have some modern support for the timing of light and dark layer deposition.

Is anything know about the oceanic/atmospheric circulation at this location? These are presumably relevant to the mechanism.

CDW's potential role seems worth exploring further especially because part of the mechanism described in Riechelsson et al is that UCDW upwelling increases when TSI increases. Warmer water into the inlet could prompt sea ice breakup.

It is also possible that changes in CDW intrusion would affect the diatom carbon isotopes. Lei et al., 2024 (Frontiers | Impact of circumpolar deep water on organic carbon isotopes and ice-rafted debris in West Antarctic: a case study in the Amundsen Sea) find that in the Amundsen Sea, increased CDW resulted in a negative shift in the carbon isotope values.

It is also worth considering changes to atmospheric circulation. Previous work demonstrates that solar cycles are amplified in the atmosphere and propagated through the climate system by shifts in the ITCZ and SWW. How might this rearrange circulation over Antarctica? Even if this isn't the cause, it is worth looking into to rule it out.

The definition and interpretation of light and dark layers is a little confusing. From visually looking at the cores in Figure 2 or even S2, it does not look like there is equal representation of dark and light intervals. Is it just light and dark or is there a third sediment type? If so, please indicate all of this in the figure. Assuming it is just the two sediment types, if the dark layers are deposited in early summer during inlet opening and the lighter layers are deposited later in the summer that would imply that the layers come from the same melt event. There is no indication in the text of layer deposition in years that the inlet does not open. That would mean the record is not continuous and data would be clustered in years where the inlet opens, with temporal gaps when it doesn't. A record that is not continuous and has higher sedimentation within specific years would influence the frequency analysis and is a cause for concern (if my understanding of this is incorrect, please clarify within the text). With this in mind, there are other ways of presenting the data that may even be more useful to the interpretation. For example, plotting TSI over time, and on that chart place dots to indicate years when a layer couplet (or bright layer) is deposited. This had the added benefit of identifying the direction of the relationship between TSI and inlet opening which can help with the mechanism (does it open when TSI increases or decreases? Does this align with the modern from satellite images?)

For a frequency analysis, the age model decisions need to be well supported. Some issues with the current age model:

1. The age/depth plot (Figure S1) is forced through 0, yet there is not a data point to indicate that the coretop is modern. The top of the core in HLF17 is dated with 210Pb. Can the brightness from both cores be plotted together to see if the two sites align?

i. Forcing the plot through 0 also causes a change in sedimentation rate in the top of the core. If these ages are real, what process would have caused this change?

2. The delta R may be consistent across time at this location, but that does not mean it is consistent across archive. The 600 delta R value was obtained through a coral study, yet none of the proxies used were corals. For the benthic foraminifera, there is no species listed, yet there could be an expected difference in reservoir age between an infaunal and epifaunal species. There could also be a difference between those species and Echinoidea, for example. This is a shallow core, so it may be the case that the same reservoir age can be used, but this choice would need to be justified with literature.

3. The lowest error on radiocarbon age is at the deepest core depth which is highly unusual.

As a suggestion, Mezgec 2017 has a sea ice (Na+) record, could this be plotted in some way with the IPSO25 to support its

use as sea ice proxy?

There should be more exploration into mechanistically how the solar cycle is propagated through the climate system to translate into sea ice extent

In regards to Figure 6, the actual difference in TSI between a max and min of a Gleissberg cycle is 0.17 W/m². This model is forced by a change of ~17 W/m² which is orders of magnitude larger. That allows for much more of a direct effect of shortwave radiation on the region, rather than signal amplification and transport through circulation as most other solar cycle studies would suggest. Instead, there should be more focus on figuring out why such a small direct signal would have the effect it does on inlet opening.

How do these results compare to models that look at changing TSI such as Swingedouw et al., 2010 which found that Antarctic temperatures are expected to cool during peak TSI in solar cycles?

If the takeaway from the paper is that changing TSI associated with solar cycles influences fast-ice extent, then the portion about IPO and AMO should be shortened/de-emphasized. It can be mentioned that the 30-year signal's presence could be the result of those processes to explain its presence, but expanding upon them diverts from the paper's purpose.

Why is the Gleissberg cycle so prominent in the fast ice record if it is so weak in the Steinhilber TSI record where the 200 year cycle is stronger? As a side question: in creating the spectra, was the entire Steinhilber record used or just the time interval overlapping with TR-17-08?

Minor General Comments:

1. In explaining the importance of fast-ice, I would be interested to see how the fast ice and total sea ice extent are related. What is the evidence that IPSO25 is fast-ice specific?
2. Figure 3: It would look a lot cleaner and convey the cycles better to only show the moving average. There is so much noise. That way the y-axis can be constrained and the trends will be more visible.
3. Figure 4: The axis colors are very confusing. Why are the highest and lowest values represented by the same color? It makes it unclear if the light lamina have very high or very low *C. pennatum* biovolume.
4. Figure 1d- The labels are much too small

I am excited to see the next iteration of this paper and what the authors are able to find!

Reviewer #3

(Remarks to the Author)

The manuscript by Tesi and colleagues presents a reconstruction of fast ice in the northwestern Ross Sea, Antarctica, over the past 3,700 years. Most previous studies have focused on pack ice, or simply sea ice, as distinguishing between pack and fast ice dynamics during the Holocene and glacial–interglacial timescales is particularly challenging. Consequently, much less is known about past fast ice dynamics, despite its significant influence on coastal ecosystems, marine-terminating glaciers, and ocean circulation. Placing current and future changes within a natural context is therefore highly commendable.

The manuscript is well-written and well-illustrated, with clear objectives and sufficient background information.

However, I have several important concerns regarding the age model and modelling approach, which must be addressed before further consideration. I hope these comments will contribute to improving what is undoubtedly an interesting study.

Major Concerns and Recommendations

Representativeness of the Study Area

The Edisto Inlet is a remote fjord that is partially isolated from global ocean circulation. While I do not question that the proposed proxies enable the reconstruction of fast ice dynamics within this inlet, I am concerned that the reconstructions may be highly site-specific, reflecting a regional rather than a broader response to global forcing due to the physiographic setting (i.e., a small fjord with two marine-terminating glaciers and no clear evidence of modified Circumpolar Deep Water (mCDW) intrusion).

One example is the differing interpretations of the dark and light laminae preserved in the studied core. In this study, dark laminae are interpreted as indicative of early summer sea ice melting (associated with sea ice diatoms and terrigenous supply), whereas light laminae are associated with summer ice-free conditions (linked to open ocean diatom production and rapid settling). However, this interpretation is opposite to findings from Wilkes Land, where dark laminae were deposited during the summer open ocean season, reflecting lower productivity and hyperpycnal plume input (Maddison et al., 2006, 2012; Denis et al., 2012). Although the laminae appear to be well interpreted in this study based on discrete sampling, the use of thin sections would undoubtedly provide further validation.

Additionally, there is no mention of the potential influence of the El Niño–Southern Oscillation (ENSO), which has been shown to exert a strong control on coastal conditions at multi-decadal timescales throughout the Holocene (Shevenell et al., 2011; Pike et al., 2013; Crosta et al., 2020; Johnson et al., 2021, and references therein).

Overall, I recommend strengthening the justification for the significance of this region, study, and interpretation to ensure greater alignment with the broad readership of Nature Communications.

Age Model and Its Impact on Results and Interpretations

The age model has been developed using a state-of-the-art Bayesian approach applied to radiocarbon dates obtained from carbonate material, which is commendable. However, several aspects of the age model warrant further discussion. The age-depth relationship appears to be constant between radiocarbon dates, which does not align with the highly laminated sediment structure. The authors state that the sharp, bright layers represent seasonal, short-lived events, whereas

the more diffuse dark layers correspond to multiple years of accumulation. This suggests that sedimentation rates vary considerably between bright and dark layers, which would significantly impact the age distribution between control points. For instance, bright laminae can be several centimetres thick (Figure 2), representing just a few weeks of sedimentation, whereas dark laminae may correspond to several years of accumulation.

Additionally, it is unclear how years with no sedimentation, when sea ice did not open (lines 119–120), have been accounted for in the model.

I was also unable to find any information on how the core-top was dated. Was the sediment-water interface preserved, and was it set at the year of collection? Clarifying this would help assess the accuracy of the most recent age constraints.

Moreover, the uncertainty in the age model increases with depth, ranging from ± 50 to ± 300 years, which represents up to 10% of the total time span covered by the core. Given the floating ages between radiocarbon dates due to highly variable sedimentation rates, as well as the inherent uncertainties of radiocarbon dating, I question the robustness of the age assignment for individual laminae. This, in turn, raises concerns about the reliability of the spectral analyses and their interpretations, particularly regarding the identification of solar cycles.

I recommend further discussion on these points to reinforce confidence in the chronology and its implications for the study's key findings.

Spectral Analyses: Approach and Interpretations

The use of REDFIT, a stationary approach applied to unevenly spaced samples, has identified significant cyclicities (~90 years and ~210 years) above the 95% confidence interval. These periodicities align with similar cycles observed in the Total Solar Irradiance (TSI) record. However, the reliability of the spectral analysis is highly dependent on the robustness of the age model, which, as outlined in my previous comments, remains questionable due to uncertainties in sedimentation rates and floating radiocarbon ages.

Given these concerns, I recommend incorporating a non-stationary approach to assess how these cyclicities evolved over the considered period. This would provide a more nuanced interpretation of the temporal variability in climate-forcing mechanisms and strengthen the study's conclusions.

Solar Irradiance and Model Output

The manuscript states in Methods (Section 4.6) and the Supplementary Information (Section 4) that pre-industrial (PI) conditions were perturbed with an abrupt 4% increase in the solar constant lasting 150 years. However, this perturbation appears to be well outside the natural range of variability, raising concerns about the robustness of the model output and its associated interpretations.

Magnitude of the Perturbation: It is unclear what the 4% increase refers to. If it applies to Total Solar Irradiance (TSI), this would correspond to an increase of approximately 50 W/m^2 from the mean PI TSI value of $\sim 1364 \text{ W/m}^2$. This far exceeds known variations in the solar constant, which are approximately 1 W/m^2 for the 11-year Schwabe cycle, and $4\text{--}5 \text{ W/m}^2$ for the 200-year Suess cycle (Dean, 2000; Schmutz, 2021).

For context, the reduction in TSI during the Maunder Minimum was only $\sim 0.2\%$, and current research suggests that the Little Ice Age (LIA) was primarily driven by volcanic eruptions rather than changes in solar irradiance (McGregor et al., 2015). Changes in solar irradiance over the Holocene have also been relatively minor, reaching only a few W/m^2 (Vieira et al., 2011).

Forcing Duration and Natural Cyclicity: The application of a 150-year constant forcing is inconsistent with natural solar cycles. For example, if considering the Suess cycle, irradiance varies from minimum ($\sim 1364 \text{ W/m}^2$) to maximum ($\sim 1366 \text{ W/m}^2$) over ~ 100 years. The imposed perturbation thus lacks a realistic temporal evolution.

Model Response to Forcing: It is also unclear why a constant 4% increase in the solar constant resulted in a continuous increase in surface shortwave (SW) radiation over the modelled 150 years. The magnitude of this increase ($\sim 20 \text{ W/m}^2$, Figure 6a) is substantial compared to changes in astronomical insolation. For instance, summer insolation at 60°S increased by only $\sim 10 \text{ W/m}^2$ throughout the entire Holocene (Berger & Loutre, 1991). This excessive forcing likely explains the significant rise in sea surface temperature (SST) by 1.5°C in the Ross Sea region (Figure 6a). A more reasonable and geophysically realistic perturbation would have resulted in a more tempered response.

Implications for Fast-Ice Dynamics: This disproportionate forcing may also contribute to the apparent contradiction between modern observations and the model output. Specifically, in situ observations suggest that atmospheric temperature plays only a minor role in fast-ice melt across Antarctica, whereas mechanical breakup—driven by wind and waves—is the dominant factor, particularly when pack ice no longer provides protection (Lines 241–244). The model, however, appears to attribute significant fast-ice loss to atmospheric warming, possibly due to the exaggerated solar forcing.

Model Resolution and Applicability to Edisto Inlet: Given that Edisto Inlet is a narrow fjord, it is important to confirm whether the model resolution is sufficient to accurately capture local processes. If the spatial resolution is too coarse, key dynamics—such as interactions between fast ice, ocean currents, and atmospheric forcing—may not be well resolved, leading to potential misinterpretations of the model results.

Recommendations: (1) Clarify what the 4% increase in solar constant refers to and ensure it falls within geophysically plausible values; (2) Consider using a more realistic time-dependent solar forcing aligned with known solar cycles; (3) Provide further discussion on why the imposed forcing results in a continuous increase in SW radiation and whether this is an artefact of model setup; (4) Address the potential contradiction between modern observations and modelled fast-ice dynamics; (5) Ensure that model resolution is adequate for capturing processes within the narrow fjord setting of Edisto Inlet. These refinements would help improve the credibility of the model results and their broader implications for Antarctic fast-ice variability.

Minor concerns and recommendations

References should be cited at their appropriate places within a sentence, rather than grouped together at the end. For example, references [4–10] in lines 54–56 should be assigned to their respective processes. Please ensure this is done

consistently throughout the manuscript.

The method for sampling paired laminae remains unclear, both in this study and in Tesi et al. (2021). Specifically : How were the paired laminae selected? What was their thickness? Were sub-laminae present, and if so, how were they accounted for? A clearer explanation of the sampling strategy would enhance the reproducibility of the study.

The relevance of diatom biovolume for reconstructing sea ice vs open ocean conditions is questionable. While biovolume is important from physiological and geochemical perspectives, absolute or relative species abundances are more relevant for assessing past environmental conditions.

Additionally, calculating biovolume introduces significant errors if diatom biometry was not performed. The size of specimens within a single sample, as well as between samples, can vary considerably. For example, *Fragilariopsis curta* frustules range from 10 to 55 μm in a single sample, with mean lengths varying between 21 and 27 μm over the Holocene (Crosta, 2009). The size variations are likely even more pronounced for the tubular *Corethron pennatum*. If biovolume estimates were based on the assumption that larger valves correspond to longer tubes with more girdle bands, it is important to acknowledge whether this relationship has been directly validated.

Given these uncertainties, it may be more appropriate to rely on species abundances rather than biovolume for environmental reconstructions.

The environmental interpretation of the light and dark laminae in this study differs from other coastal areas, where light laminae typically represent spring melting, and dark laminae indicate summer ice-free conditions with reduced biogenic sedimentation and increased terrigenous input from hyperpycnal plumes. This contrast should be explicitly mentioned in the manuscript.

Additionally, a thin-section analysis would be highly beneficial for verifying the interpretation of the laminae structure and depositional processes.

In accordance with international taxonomic rules, genus names must be fully spelled out when starting a sentence (e.g., Line 141). Please ensure this is corrected throughout the manuscript.

Lines 161-163. The current wording suggests a strict seasonal signal in the laminae, which contradicts earlier statements. A clearer approach would be to describe dark laminae as indicative of years with extensive sea ice cover. Light laminae as deposited occasionally and rapidly during years when the inlet opens up. This explanation is better articulated in other sections of the manuscript and should be consistently applied.

The band-pass filter analysis does not seem to add significant value, as it naturally aligns with the original grey-scale record given the spectral analysis results (90- and 210-year cyclicities). The more valuable insight comes from identifying offsets between the raw and filtered records, indicating non-stationarity in the cyclicities. Instead of a band-pass filter, a wavelet analysis would be more appropriate for capturing temporal variations in these cycles.

Lines 211-212. The sentence regarding the impact of positive SAM (SAM+) is unclear. Generally, a positive SAM strengthens Ekman transport northward, leading to increased sea ice extent and decreased coastal concentration. This seems to contradict the statement in the manuscript. Or are you suggesting that SAM+ enhances sea ice transport from the southern Ross Sea and accumulates it around Cape Hallett? If so, this should be explicitly stated.

The section Lines 236-260 is excessively long and does not contribute significantly to the study's core findings. A more concise version would improve readability and focus.

Lines 355-356. The phrase "one distinct specimen" is ambiguous—does this refer to one frustule, or one whole diatom, which consists of two frustules? If the latter was not accounted for, the abundance of *Corethron pennatum* may have been overestimated by a factor of two. Please clarify.

Sections 4.6 in the Methods and section 4 in the SOM are nearly identical, except for Table S2. Either consolidate them or clarify their distinct purposes to avoid unnecessary repetition

Figure 2: Some sections (e.g., I, VIII, XII) appear massive, lacking clear lamination. Could this be an issue with the photographs, or is it representative of the actual sediment structure? If it is real, a brief explanation of why these sections appear massive would be useful.

Figure 4: The current colour scheme is confusing, as yellowish tones appear at both ends of the spectrum. This makes it difficult to distinguish between low and high values. Please replace the very low *Corethron pennatum* biovolume category with purple to create a more intuitive gradient.

Lines 47-48 of the SOM: The presence of laminae with distinct depositional timeframes (dark = multi-year accumulation, bright = short episodic events) contradicts the statement that "accumulation rates vary smoothly downcore." Please rephrase this section to reflect the variability in sedimentation rates due to the laminated structure.

Table S1: The age of the core top is missing. Please report this value and explain how it was determined. Specifically, was the sediment-water interface preserved during core retrieval? What method was used for age assignment (e.g., ^{210}Pb dating, radiocarbon dating, or another approach)?

I hope these comments and recommendation will prove useful.

Cordially.

Xavier Crosta

Reviewer #4

(Remarks to the Author)

Version 1:

Reviewer comments:

Reviewer #1

(Remarks to the Author)

General comments:

We appreciate the authors' attention to detail and consideration of all the comments from our initial review. This version represents a significant improvement. The figure quality is much improved and this improves the clarity of the manuscript.

Major Comments:

Line 520: The sea ice data analysis methods section, Section 4.7, does not include how landfast ice within Edisto Inlet was defined and differentiated from mobile sea ice within the inlet. If the authors are defining all pixels which were identified as ice, based on the ice surface temperature, within Edisto Inlet as landfast ice we do not find this convincing enough evidence to be called landfast and not just sea ice or ice covered. To be defined as landfast there needs to be a component of immobility over a period of time in addition to being identified as sea ice. Or is it based on the diatom assemblages commonly associated with fast ice? At a minimum, this caveat should be acknowledged in the manuscript.

Minor Comments:

Line 75-77: the sentence ... with East Antarctic and Bellingshausen Sea exhibiting the largest positive trend (about 1.1% yr⁻¹)... Victoria and Oates land (-1.8% yr⁻¹) is not entirely accurate and slightly misleading. From Fraser et al. (2021) 1) there is no East Antarctica region, but I am assuming the authors meant the Australia region. 2) by the percentages the region with the largest positive trend is the Bellingshausen Sea (2.8 %yr⁻¹) and largest negative trend Weddell Sea (~-2.6 %yr⁻¹). If you are saying a region is the largest and then use percentages, the regions with the largest percent should be used.

Line 58: "bottom waters" to "bottom water"

Line 114: This should refer to Figure 1a not just Figure 1

Line 126: Instead of referring to the "Methods" section this sentence should refer to the section number (i.e. Section 4.2 for the age depth model).

Line 215: a reference to Figure 1a and the yellow polygon denoting the region of northern Victoria Land where the pack ice concentration was calculated would improve the clarity

Line 233: From Figure 7 there seems to be a lagged response of ~15 days from when the pack ice concentration decreases to the ice-free inlet % going up. This is speculative and based on the visuals of Figure 7. However, if there is a consistent lagged relationship this would provide grounds for some speculation and discussion on the physical link between pack ice concentration and the ice-free Edisto Inlet.

Line 368: This is merely a suggestion and realize that we as the reviewer should not dictate the content of another's paper. Since the main point of this paper is the ability to say if Edisto Inlet was ice free or not for the past ~3.7 thousand years we feel like there is a figure missing that displays a general idea of the trend of ice covered or not over the period of the core. We envision something like a climate stripes figure (examples can be found <https://www.reading.ac.uk/planet/climate-resources/climate-stripes>). For this manuscript the lines could be binary indicating if Edisto Inlet was ice covered or not during each year or each stripe could be a percentage of a 10 year where hues of blue indicate more ice covered years and hues of reds indicate more ice free years for example. The reviewers believe a figure like this would provide visual evidence for the periodicity of ice covered, vs not, throughout the time represented by the core. Without a figure, and accompanying text, displaying the ice conditions of Edisto Inlet over the period of the core we feel a large and important part of this paper is missing.

Line 536: Solp4p needs to be capitalized.

Line 555: Figure 1b. This figure should be more zoomed in on Edisto Inlet such that a reader could differentiate between the 3 points. Cape Hallett should also be annotated on Fig 1b, since it is referred to in the text.

Line 659: No need to refer to Figure 1b here. Could be reworded to the percentage of Edisto Inlet ice free.

Line 609: Is there a high pass filter line within Figure 3a? If so, it should not be black. If not, our apologies, as the print makes it look like there is a black high pass filter line.

Paragraph beginning line 270, plus throughout the discussion: It is hypothesized a few times that solar absorption in an adjacent non ice-covered ocean (e.g., if pack ice has retreated) can weaken fast ice and eventually lead to its melt or breakout. There is actually a good reference for this hypothesis: <https://doi.org/10.5194/tc-14-2775-2020> - which is currently

not included in the reference list. I think it could strengthen this argument.

General for Figures: The way the author denotes which axis corresponds to various lines within a plot changes. Adopting a consistent format is suggested. For example: Figure 5a/b the axis tick colours denote which axis is associated with which line while figure 6b/c the axis label colour denotes that the right axis is associated with the red line. List of figures with inconsistent notation of associated axis: Figure 5a/b, 6b/c, 7a-f, 8d.

Reviewer #3

(Remarks to the Author)

Tesi and co-authors have provided an insightful rebuttal letter and a thoroughly improved revised version of the manuscript. I commend the considerable amount of work undertaken to address the reviewers' comments. I particularly appreciated the addition of modern data demonstrating that sea ice conditions in the inlet are of regional relevance and that the proxies reflect different aspects of sea ice variability. The revisions have addressed most of my initial concerns; however, two main issues remain.

Firstly, I still believe that the band-pass analysis is not appropriate, as such methods tend to reveal cycles regardless of their significance. The fact that the amplitude of the cycles varies over time suggests they are non-stationary. This is further supported by the ~90-year cycle falling below the 95% confidence interval. I therefore recommend replacing the band-pass and spectral analyses with a wavelet analysis, which accounts for non-stationarity and would be far more suitable. I raised this point in the first round of review, but did not receive a satisfactory response. The revised spectral analysis, which incorporates age uncertainties, still does not address the non-stationarity of the cycles.

Secondly, I remain concerned about the discrepancy between the primary interpretations drawn from the data—highlighting winds as the main driver of sea-ice variability in the northern Ross Sea and the inlet—and those based on the modelling results, which suggest temperature as the dominant factor. This contradiction is only briefly acknowledged in the main text. The imposed 4% increase in solar forcing is clearly beyond observed TSI variability, and this should be explicitly stated. I frequently noted that the interpretations derived from the model contrast with those based on the empirical data (see the annotated file). The added value of the model must be more clearly articulated and substantiated. I also question—though this is outside my area of expertise—whether more climatically relevant model experiments exist, for example those using more realistic solar forcing scenarios.

The annotated file includes further minor comments.

I hope these suggestions will help enhance what is already an engaging and valuable study, likely to be of considerable interest to the broad readership of Nature Communications.

Reviewer #4

(Remarks to the Author)

Version 2:

Reviewer comments:

Reviewer #1

(Remarks to the Author)

Thanks again to the authors for addressing all our comments. We look forward to citing this paper.

Reviewer #3

(Remarks to the Author)

The second revised version adequately addresses most of the reviewers' comments or provides convincing rebuttals. The manuscript is exhaustive, well-supported, well-written, and well-structured, and I believe it is now ready for publication.

Cordially,

Xavier Crosta

Reviewer #4

(Remarks to the Author)

FOR ALL REVIEWERS:

We present a substantially revised manuscript that includes new data and additional material.

What's new:

- 1-y data from a sediment trap deployed inside Edisto Inlet
- ²¹⁰Pb data from a short core (multicore) collected in the piston coring region
- A satellite image time series analysis of fast ice and pack ice (MODIS)
- A new spectral analysis that accounts for propagated age uncertainties

REVIEWER COMMENTS

Reviewer #1 (Remarks to the Author):

Together with an ECR, as encouraged by NComms, we have now finished a review of NCOMMS-24-83194. As required, we together submit this single report incorporating both of our views on the manuscript.

This manuscript represents an incredible amount of rigorous work - including a marine science voyage, sediment core extraction, sediment dating, rigorous work defining layers within the sediment, diatom analysis to determine overlying ice conditions, spectral analysis of the record, and comparison with CMIP6 data simulating a different solar constant.

While the details of the sediment work are a little out of our area of expertise, we found no major problem with what was presented in that regard. We did, however, find some major shortcomings with the methodology which aims to provide evidence of the insolation cycles being the primary driver of fast-ice stability, and these are outlined below.

Major Comments

1) There are time series presented of band-passed insolation and sediment brightness (Fig 5 b and c), and it is argued that insolation modulates the sediment greyscale (and thus fast ice). However, no correlation is performed between these. Without such an obvious correlation being performed, the reviewer gets sceptical that the correlation is low and/or insignificant. If

that were to be the case, then the entire causality argument hangs on an argument that the periodicity is similar, so it must be the driver. This is not really strong enough evidence to make this claim. The periodicity isn't even a match for the middle period discussed (90 vs 110 years), and for that band the 110 year period isn't even very prominent. I may be wrong, and the correlation might prove significant, but eyeballing Fig 5c, it doesn't look promising. Fig 5b maybe but it's too hard to eyeball it (plus there's a spectrum mismatch for this band, as noted above). Basically it's not a strong argument without a correlation. And the argument is the crux of the manuscript (it's in the title).

We appreciate the reviewer's concern regarding the absence of a direct correlation analysis between insolation and sediment brightness. However, in sedimentary archives, and more broadly in paleoclimate records such as corals, tree rings, and ice cores, one-to-one correlations with solar cycles are rarely resolved due to two primary reasons:

First, all climate archives are subject to inherent age uncertainties in their chronological models. These uncertainties make precise, calendar-year alignment between proxy records and external forcings (e.g., solar variability) inherently difficult. Among various climate archives, marine sediments often exhibit the largest age uncertainties, primarily due to variable and often non-linear sedimentation rates. Only annually laminated (varved) records can achieve the precision required for direct temporal correlations.

Second, in archives like tree rings or ice cores, it is sometimes possible to use an internally derived proxy for solar activity (e.g., radiocarbon-14C or beryllium-10), which is subject to the same age model as the climate signal itself. This reduces the problem of temporal misalignment. In marine sediment records, such internal solar proxies are typically unavailable. Consequently, a well-established alternative approach in the literature is to examine frequency components in the proxy record rather than relying on direct time-domain alignment.

This is the approach we adopted in our study.

We acknowledge and appreciate that both Reviewer #2 and Reviewer #3 emphasized the need to account for how age model uncertainties may influence spectral analysis. In response, we have revised our methodology to include a Monte Carlo-based sensitivity test. Specifically, we generated 1,000 plausible age-depth scenarios and computed the power spectrum for each realization. We then assessed the robustness of spectral peaks against a red-noise background, following the approach of Schulz and Mudelsee (2002)¹. This updated and more statistically rigorous method confirms the presence of two distinct low-frequency bands, consistent with our original findings.

We recognize that, from the reviewer's perspective, the connection to solar variability may seem speculative. However, it is important to emphasize the broader structure and intent of our paper, which aims to extend the temporal boundaries of modern observations. The manuscript follows a logical progression:

1. We demonstrate that sediment color reflects different thawing mechanisms, showing that marine sediments can effectively expand satellite-based observations.
2. We leverage image analysis to significantly increase data density, thereby improving the temporal resolution of our record.
3. Our fast-ice record reveals two dominant low-frequency bands. The recurrence of white laminae during periods of inlet opening is therefore not random, but suggests underlying periodic forcing.
4. While not conclusive, we investigate whether solar cycles could be one such driver. To support this, we incorporate new lines of evidence, including literature-based re-analyses and complementary records from continental archives (e.g., ice cores), which independently suggest links between solar variability and atmospheric circulation patterns.

In light of the reviewer's feedback, we have clarified the methodological framework and rationale in the revised manuscript. To further reflect this shift in emphasis, we have also modified the title to remove explicit reference to "solar cycles," instead highlighting the novel methodological approach and its potential to extend the observational record beyond the satellite era.

2) Similarly, it's stated a couple of times that the shorter periodicity is due to IPO or AMO variability. There's even less evidence presented for this (no analysis at all, just a hypothesis), but it's upheld as fact in places (e.g., lines 169, 299). This is very much overstated and should be heavily caveated.

The revised spectral analysis does not show significant evidence for high-frequency periodicities, suggesting that if such signals were present, they were likely not prominent or persistent. It is also important to note that our updated spectral technique involved segmenting the greyscale data into six equally long, overlapping intervals, calculating individual spectra for each segment, and then averaging them. While this approach enhances the robustness of the spectral estimate, especially in the presence of age uncertainties, it also tends to smooth the spectrum and suppress less consistent high-frequency components, further reducing the visibility of transient or noisy signals.

L264-277: The discussion on mechanism here is lacking in evidence. Katabatic wind representation in CMIP class models is poor and unreliable. There's also no evidence given to show that downwelling shortwave is the primary factor for the reduced pack-ice extent. Regarding the comment on snow cover, Ushio (2006) showed that a deep snow cover favours stability, i.e., in the opposite sense to the hypothesis given here. This lack of evidence is summed up in the first sentence of the next paragraph (L279). The CMIP simulations do provide evidence that the surrounding environment changes with +4% insolation, but the links to fast ice are too nebulous, without any correlation presented between insolation and the sediment colour.

In this section, our aim was to present a range of processes that may influence fast-ice breakup. Regarding katabatic winds, we acknowledge that our intention may have been misunderstood. We did not aim to establish a direct link between katabatic winds and our model outputs. Rather, the cited reference is a review paper that discusses various regional influences, including katabatic winds, as part of a broader context. Our primary focus in this part of the study is on regional meridional and zonal wind patterns as simulated in the AbruptSol experiment, not specifically on katabatic winds.

We recognize that the structure of the original manuscript may have made this unclear, and we appreciate the reviewer's comments for highlighting this point. In the revised version, we have restructured this section to improve clarity and flow, making our emphasis on large-scale regional wind patterns more explicit.

This final section of the manuscript builds upon earlier results, first, by establishing sediment color as a reliable proxy for thawing mechanisms, and second, by identifying two persistent low-frequency bands in the spectral analysis. We then explore potential drivers of these cycles, including solar variability, with the goal of identifying plausible mechanistic links. While we do not claim definitive proof of a single causal mechanism, we believe it is scientifically reasonable to propose informed hypotheses supported by multiple lines of evidence. In this revised version, we have expanded our discussion to consider a wider range of potential mechanisms, drawing from both reanalysis data and complementary continental climate archives (e.g., ice cores), in order to contextualize our marine sediment record within a broader climatic framework.

We also thank the reviewer for identifying the incorrect citation of Ushio (2006)². We have corrected this in the revised manuscript. Regarding the insulating effects of snow cover, we have included a more nuanced discussion focusing on studies by Massom et al (2001)³ and Sturm and Massom (2017)⁴.

Figure quality is questionable - all except fig 4 display compression artefacts. Fig 1 is particularly poor. Inset c has no scale and includes illegible text. The “satellite image” has no more information on which satellite. It shows fast ice covering the region in January, which is a bit misleading (this region has fast ice breaking out most summers). Panel d has illegible axes, and somehow two x-axis labels which aren’t explained. The right scale has no label. The caption confuses the orange and green dots (I think they should be switched in the caption). “This study” is ambiguous in the caption. The “red line” is almost not red due to compression. Fig 2 and 3 are fine aside from the compression of text. Fig 4 has a poor choice of colour bar (is yellow 20 or 100?) making interpretation of that very difficult. Fig 5, as mentioned above, suffers from poorly-labelled axes, often missing units. The “highpass” is interesting and completely ignored in the method - why was a high pass needed? In panel a why was the 30 year period never annotated? Fig 6a is very confusing. What is “year”? Year since the start of the AbruptSol experiment? It’s very unclear. The orange label is clearly wrong - I guess “increase vs control” is needed (insolation should be over 100 W/m² even averaging over a cloudy year)? Is this averaged over a year? PI is not defined in the comment. There seems to be interchangeable use of “solp4p” and “AbruptSol” - can’t these be combined? Panel c’s colour bar doesn’t give the “sense” (e.g., red is northward) but I concede it’s possible to figure this out by carefully reading the caption - but why not make it easier for the reader? I paid less attention to the supp figures, but S2 is really illegible when printed. The purples get completely lost and the blue isn’t sharp at all.

We would like to note that the PDF generated during submission uses low-resolution images for practical purposes. In response to the comments, we have updated the figures, including those in the supplementary material. For example, we increased the font size in Figure S2 and changed the color in Figure S3 from purple to pink. In general, most of the figures have been updated in both manuscript and supplementary materials.

Some of these aspects are described in greater detail below in the “minor comments” section.

Minor Comments

Line 1 - coast is a descriptor of the location not part of the name thus should not be capitalized

Changed accordingly

Line 27 - “Fast-ice” should not be hyphenated in this scenario. In this scenario “fast ice” is a noun and should not have a hyphen. “Fast-ice” should be used as an adjective ex. fast-ice extent. This problem persists throughout the document

Changed throughout the manuscript

Line 32 - Antarctica is misspelled

Changed accordingly

Line 32 - The use of the oxford comma is inconsistent throughout the document

Checked throughout the manuscript

Line 32 - “image analysis at high resolution” does not match the description later in the document of a high resolution image analysis.

We have replaced this definition in the revised version

Line 34 - solar activity not defined (sunspots?), and CMIP6 not properly defined.

We now provide a definition of solar activity and CMIP is properly introduced

Line 55 - “mixed” is normally “modified” and “waters” is normally “water” here.

Changed accordingly

Line 58 - fast ice has been shown to modulate glacier tongues, not tidewater glaciers.

Not sure if this is a problem of definition but we were referring to the tongues (floating part) of tidewater glaciers

Line 59 - not seal breeding - just foraging (as far as I know).

Corrected accordingly

Line 70 - I don't agree with “highly approximated” here - the fast ice simulations are quite realistic, but the downside is that they are only regional now.

Changed to highlight the regional limitation

Line 71 - referencing the wrong Fraser paper (should be 2020 for the dataset)

Corrected

Line 80 - show-case should not be hyphenated used showcase

Changed

Line 83 - “consolidated” isn’t enough of a description. Actually, this fast ice breaks out most years in recent times, and this isn’t really stated clearly anywhere.

“Removed the word consolidated”

Line 97 - The introduction does not include descriptions of the Gleissberg and Suess-de Vries solar cycles. The introduction would benefit from information about these solar cycles given a lot of the following information and conclusions rely on an understanding of what these cycles mean and their time scales.

We prefer to keep the introduction concise, which is already quite long due to the additions requested by the other reviewers. Given the shift in focus, we would prefer to omit this unless it is strictly necessary.

Line 108 -109 - Missing error value “3.7 ± ka BP”. There should be a value after the ± to establish the time error.

Corrected

Line 133 -155 - The content is synthesis of the results and better fits in a discussion section
This part has been removed from the first draft

Line 150 - “more or less” is ambiguous.

Edited

Line 157- “fast sea ice” is new terminology not used elsewhere.

It was a typo, corrected

Line 158 - Edisto needs to be Edisto Inlet to improve clarity and allow the reader to reference themselves within Figure 4

Corrected throughout the text. Not sure what “reference themselves within Figure 4” means

Line 165 - should be “deconvolve”

Corrected

Line 198 - "Overall, although" is awkward phrasing

Corrected

Line 200 - 204 - Figure 6a, which is what this text is describing, displays increasing surface shortwave radiation, SST, and precipitation which coincide with a decrease in sea ice area fraction. There are connections made between the variables which are logical but no evidence that they have any dependence or affect on each other is provided. This can be done in the form of citations which found the connection or statistics done on this data.

In the revised manuscript, we have reorganized the relevant section to better explain the complex interacting processes involved in sea ice change over the Ross Sea. We have also appropriate citations to support the connections between the variables.

Holland et al. (2017)⁵ used observations and climate model outputs to highlight a complex, seasonally varying relationship between winds, surface shortwave radiation, and sea ice changes over the Ross Sea under present-day climate conditions. Their results showed that stronger westerly winds in austral spring results in export and dynamical thinning of sea ice in the western Ross Sea. This leads to earlier sea ice retreat and longer ice-free season, which increases absorption of shortwave radiation during summer. Consequently, anomalously warm summer surface ocean temperatures delay sea ice formation, resulting in reduced sea ice concentration in March and subsequent months.

The role of solar forcing was investigated by Stewart et al. (2019)⁶ who demonstrate that solar activity has significant implications for Antarctic sea ice dynamics, particularly through its influence on polynya formation and heat transfer processes and, in turn, ice shelf stability. Their observations reveal that seasonal reductions in sea ice, driven in part by wind-driven export, expose open-water regions such as the Ross Sea Polynya to increased solar radiation. This leads to substantial surface ocean warming, which delays refreezing and enables the formation of a warm, buoyant surface layer. This solar-heated layer can intrude beneath adjacent ice shelves, enhancing basal melt rates by an order of magnitude relative to the shelf-wide average. The study highlights a feedback mechanism wherein diminished sea-ice cover promotes increased solar heat absorption, further suppressing ice formation and enhancing ocean heat content.

All of these points, along with additional supporting references, have been incorporated into the revised manuscript.

Line 211 - 212 - this reference indicates sea ice extent increases by a positive SAM, not a concentration

We have revised the text accordingly

Line 216 - “wind-driven sea ice loss is negligible” is not shown anywhere.

This means that changes in sea ice do not correspond directly to changes in meridional wind.
We have revised the wording for clarity

Line 225 - This statistic is from Fraser et al. 2021 and not cited

Added

Line 226 - double use of “and” is not grammatically correct

Corrected

Line 236 - 260 - Paragraphs are introduction material not results.

This section is part of the discussion where we highlight different mechanisms, which we believe is important for interpreting and discussing our results. However, we have shortened it to improve the overall flow of the paper

Line 236 - I was expecting a citation to support the statement following made in the sentence which starts “A survey of the recent literature indicate”

The references were in the following sentence. We moved them up.

Line 264 - there appears to be a ‘that is a typo at the beginning of this line

This section is not part of the text anymore

Line 264 - 267 - This sentence is confusing and needs to be revised. Suggestion to make break into shorter sentences

This section is not part of the text anymore

Line 269 - 273 - no evidence is provided to substantiate the conclusion that increase in downwelling shortwave radiation is the primary driver of pack ice loss. SST, and precipitation also increase and no evidence was provided to quantify the importance of each driver

We agree that quantifying the relative contribution of individual drivers would strengthen the interpretation. However, the solar forcing applied in the CMIP6 AbruptSolp4p experiments affects not only surface shortwave radiation but also modifies cloud properties (optical depth, cloud height etc.), atmospheric temperature and water vapor content, and large-scale atmosphere and ocean circulation patterns. Isolating the impact of these processes would require targeted sensitivity experiments (isolating each factor individually), which are computationally intensive and beyond the scope of the current study.

In the revised manuscript we have modified the text to more cautiously reflect the limitations of the Solp4p experiments. Rather than attributing sea ice loss primarily to shortwave radiation, we now emphasize that sea ice loss is driven by multiple interconnected factors.

Line 277 - No evidence is provided to substantiate the claim the precipitation of snow is a first order effect in the context of this paper.

In the revised manuscript we have removed the precipitation results and updated the relevant discussion reflecting this.

Line 282-283 - the use of collectively and operating together mean the same thing and do not need to both be said

This section is not part of the text anymore

Line 283 - When a statement like “a survey of current literature” is made I would expect to see a citation to substantiate the second half of the sentence.

This section is not part of the text anymore

Line 291 - need to indicate if this effect applies to Antarctic air or sea surface temperatures

Arctic temperatures, text edited

Line 294 - Southern should not be capitalized as it described a location and is not the name of the location

Changed according to this comment

Line 313 - ΔR needs to be in either (ΔR) or commas as it is an abbreviation of reservoir effect and used later as a stand alone representation of reservoir effect.

Edited according to this comment

Line 319 - MCMC simulation is an abbreviation and not defined prior to being used

Edited according to this comment

Line 334 - need space between numerical value and the units

Added a space as suggested

Line 382 - Comma misplaced. Should be after “manual counts” instead of “rather subjective”

Corrected accordingly

Line 406 - solp4p should be capitalized since it is capitalized throughout the text

Corrected accordingly

Figure 1 - The orientation of 1a and 1b are inverted and there are not visual indications of this

Figure 1c - Need some form of scale

Figure 1d - Tick labels all very small and illegible

Figure 1d - left and right axis are the same, don't need both

This figure has been substantially revised: the line was removed as it appeared too small, Fig. 1c was omitted, the font size was increased, and a redundant label was eliminated

Figure 4a - Color bar wraps where values of 20 and 100 have similar colors. This is misleading and the color scale need to not wrap when the data values do not wrap

Figure 4 b and c - the x-axis values are misleading when comparing these plots. A bar with the same width represents 2 % bin of biovolume in b but 3.33% bin in c. Are the bins different sizes?

The colorbar was incorrectly exported during PDF creation; this has been corrected, and the bin size is now uniform

Figure 5 - Axis labels could be simplified

This figure has been replaced with a new one

Line 1 - coast is a descriptor of the location not part of the name thus should not be capitalized

Corrected according to this comment

Table S1 - How are you resolving a 30 year cycle for IPO when the error is between 23-91 years?

The new model accounts for age uncertainties; however, it smooths out high-frequency signals, including noise. As a result, the updated spectral power shows no strong evidence of high-frequency climate signals, except for the 90- and 240-year cycles

Figure S1 - Both x and y axis labels and tick labels need to be larger

Figure S1 - x axis needs more labels when, more frequent than ever 500 cm

Figure S1 - What are the blue spikes?

Figure was updated according to these comments. Blue symbols are the radiocarbon dates as explained in the caption.

Figure S2 - Purple lines are very hard to see with the dark image as the background

Figure S2 - the floating average displayed in S2a is significantly smoother than the floating average in b and c. Please confirm the floating average in each figure is calculated the same.

We replaced the purple lines with pink lines.

Reviewer #2 (Remarks to the Author):

The data sets presented in this paper are interesting and can provide insight into multidecadal influences on ice dynamics and possibly an opportunity to learn about regional climate connections. Deeper consideration into the mechanism around the observed variability could expand the paper's impact/relevance by connecting it to the broader setting as well as focus the paper's narrative. Additionally, for frequency analysis, decisions regarding the age model need to be better supported.

Major Comments:

More about the modern setting could help support the paleo interpretation/mechanism.

Something similar to Figure 5 in Tesi et al., 2020 could be useful in demonstrating: 1. Seasonal progression of sea ice melting and 2. Interannual variability in inlet opening. In L76-77, it is mentioned that Fraser et al has a few decades of satellite data. Could the authors compare inlet

opening in the satellite images with the TSI and/or SAM index to see the correlation (does the inlet open in years when TSI increases or decreases? Positive or negative SAM?).

We have introduced a new section focused on sea ice imagery and trend analysis, presenting the longest available time series beginning in 2000, with the advent of MODIS satellite data. However, this time span is too short to establish robust long-term correlations with Total Solar Irradiance (TSI). Additionally, anthropogenic forcing must be considered, which remains poorly understood, even for well-studied cryospheric elements like sea ice, which has shown considerable variability in regions such as the Ross Sea, possibly due to human influence as argued by the literature.

A key finding from our analysis is the interconnected behavior of pack ice and fast ice. While this relationship is not new, our results confirm it in a specific regional context. In particular, persistent pack ice in the northern Victoria Land region is associated with more stable fast ice within the inlet, which breaks up less readily. This indicates that ice dynamics within the inlet are shaped not only by local factors but also by broader regional processes.

Is there any sediment trap data at all from the broad area? It would be helpful to have some modern support for the timing of light and dark layer deposition.

We used preliminary data from a one-year sediment trap deployment in the inlet (adjacent to the coring site) to support our diagnostic tools, particularly the use of proxies such as IPSO25 and stable carbon isotopes ($\delta^{13}\text{C}$), to trace different stages of the inlet's opening. Taxonomic analysis was not performed, as this task falls under a separate project that deployed the sediment trap in the first place; however, the principal investigator kindly granted us access to the data for our study.

The new figure illustrates that toward the end of summer when the inlet is fully open for a long time, both total organic carbon (TOC) and sediment fluxes remain high, while IPSO25 concentrations are extremely low and $\delta^{13}\text{C}$ values are notably depleted, consistent with an open-water phytoplankton bloom. In contrast, during the initial fast ice breakup in the following summer, IPSO25 flux increases significantly alongside a rise in sediment flux, and $\delta^{13}\text{C}$ values become heavier, consistent with the release of sympagic diatoms. These findings provide strong evidence that our biogeochemical tools are applicable to sedimentary records and can effectively capture fast ice dynamics back in time.

Is anything known about the oceanic/atmospheric circulation at this location? These are presumably relevant to the mechanism. CDW's potential role seems worth exploring further especially because part of the mechanism described in Riechelson et al is that UCDW upwelling

increases when TSI increases. Warmer water into the inlet could prompt sea ice breakup. It is also possible that changes in CDW intrusion would affect the diatom carbon isotopes. Lei et al., 2024 (Frontiers | Impact of circumpolar deep water on organic carbon isotopes and ice-rafted debris in West Antarctic: a case study in the Amundsen Sea) find that in the Amundsen Sea, increased CDW resulted in a negative shift in the carbon isotope values.

As mentioned in the main text, there are no published studies providing evidence of Circumpolar Deep Water (CDW) intrusion into the fjords of northern Victoria Land. Even data from the mooring line equipped with temperature and salinity sensors did not indicate any such intrusion (data not shown). Although we cannot definitively rule it out, the intrusion of modified CDW (mCDW) into these fjords appears unlikely, particularly given the presence of a sill that isolates the fjords from the open ocean.

Taking into consideration that $\delta^{13}\text{C}$ can vary for different environmental conditions (shape of algae, CO₂ concentration, diagenesis, nutrients, etc), the $\delta^{13}\text{C}$ time series of the sediment trap fully support the idea that the contrasting $\delta^{13}\text{C}$ signatures are primarily driven by shifts in algal succession (open water vs sympagic algae) consistent with the literature in the Ross Sea (Munro et., 2010)⁷. Depleted $\delta^{13}\text{C}$ values are associated with typical phytoplankton that flourish under ice-free conditions, while heavier values are linked to sympagic algae, which exhibit a distinct isotopic fingerprint. The heavier $\delta^{13}\text{C}$ composition observed in sympagic algae is likely due to limited carbon availability in the constrained environment beneath sea ice. Under these conditions, carbon fractionation is reduced because the substrate becomes less abundant, leading to a lower discrimination against the heavier isotope. Although we cannot present all the data collected from the mooring line, since this dataset belongs to a different project, as previously mentioned, the available physical data do not indicate evidence of intrusion of modified Circumpolar Deep Water (mCDW). Instead, they suggest an active exchange with shelf water masses, possibly driven by the region's strong tidal dynamics.

It is also worth considering changes to atmospheric circulation. Previous work demonstrates that solar cycles are amplified in the atmosphere and propagated through the climate system by shifts in the ITCZ and SWW. How might this rearrange circulation over Antarctica? Even if this isn't the cause, it is worth looking into to rule it out.

We fully agree on the importance of atmospheric circulation and now we provide a better discussion. In fact, we delved further into the literature and found compelling evidence of atmospheric changes recorded in ice cores collected near McMurdo Station. Ice core studies, in particular, identified a strong link between aerosol composition and Total Solar Irradiance (TSI)^{8,9}. Using reanalysis data, the study suggests that TSI influences surface zonal winds during

Spring in the northern Ross Sea, which in turn affects the chemical composition of the McMurdo ice cores¹⁰. Additionally, another study demonstrates that Spring anomalies in surface zonal winds in the outer Ross Sea can lead to the removal of pack ice along the northern Victoria Land coast in the following summer⁵. An efficient and anticipated pack ice removal has the potential in turn to trigger fast ice breakup, as evidenced in our time-series analysis (Fig. X).

The definition and interpretation of light and dark layers is a little confusing. From visually looking at the cores in Figure 2 or even S2, it does not look like there is equal representation of dark and light intervals. Is it just light and dark or is there a third sediment type? If so, please indicate all of this in the figure. Assuming it is just the two sediment types, if the dark layers are deposited in early summer during inlet opening and the lighter layers are deposited later in the summer that would imply that the layers come from the same melt event.

In the supplementary material, we provide an example of layer definitions and our the script works. In a simplistic way, there is a series of dark units and a series of light units. However, visually, our eyes tend to have a bias toward higher contrast, so they are drawn to the very light intervals, which are comparatively less frequent.

First and foremost, it's important to emphasize that the number of dark versus light layers does not influence the frequency analysis, which is based on the full brightness spectrum. As for the number of intervals, while some of the white laminae are well defined and visually prominent, the coloration spans a range of tones. The software was specifically trained to detect even subtle variations in brightness. That said, the number of identified intervals is lower than the number of years represented in the record, indicating that the laminated pattern does not reflect an annual signal (i.e., these are not true varves). This interpretation is supported by the satellite image time series, which reveals a heterogeneous sequence of events, ultimately resulting in a complex and irregular laminated pattern. In parallel, our spectral analysis indicates that the opening is not random but there are cycles at low frequencies when the inlet tends to open more often.

There is no indication in the text of layer deposition in years that the inlet does not open. That would mean the record is not continuous and data would be clustered in years where the inlet opens, with temporal gaps when it doesn't. A record that is not continuous and has higher sedimentation within specific years would influence the frequency analysis and is a cause for concern (if my understanding of this is incorrect, please clarify within the text). With this in mind, there are other ways of presenting the data that may even be more useful to the interpretation. For example, plotting TSI over time, and on that chart place dots to indicate years when a layer couplet (or bright layer) is deposited. This had the added benefit of identifying the direction of the relationship between TSI and inlet opening which can help with

the mechanism (does it open when TSI increases or decreases? Does this align with the modern from satellite images?).

We analyzed ^{210}Pb data from a multicorer collected at the coring site. Our results show that, despite the laminated pattern reflecting different opening regimes, the sediment accumulation rate is relatively continuous. This does not imply that accumulation rates are constant throughout the core, but rather that the record shows no evidence of long-term interruptions or hiatuses.

The direct correlation between TSI and the laminated pattern is addressed in our response to Reviewer 1. Specifically, climate archives, especially marine sediments, carry significant age uncertainties that hinder precise alignment with external forcings like solar variability. While some archives (e.g., tree rings, ice cores) can internally trace solar signals, marine records lack such proxies, making frequency-based analysis a more reliable alternative to direct temporal comparisons. Considering the age uncertainties and other limitations highlighted above, spectral analysis is a widely used approach in TSI studies dealing with sediment archives.

That said, we also emphasize that we have downplayed the role of solar activity and instead highlight the novelty of our work: the development of a new tool that extends the temporal applicability of satellite-based fast ice reconstructions into the late Holocene. This allows for a better understanding of natural variability, especially considering that satellite records are only available for the past few decades and largely coincide with anthropogenic forcing.

For a frequency analysis, the age model decisions need to be well supported. Some issues with the current age model:

1. The age/depth plot (Figure S1) is forced through 0, yet there is not a data point to indicate that the coretop is modern. The top of the core in HLF17 is dated with ^{210}Pb . Can the brightness from both cores be plotted together to see if the two sites align?
 - i. Forcing the plot through 0 also causes a change in sedimentation rate in the top of the core. If these ages are real, what process would have caused this change?

This was a very useful comment. We conducted ^{210}Pb analysis on the long core, which confirmed that the core was missing its top section. As a result, we constructed a new age-depth model using only the top tephra layer and the lowermost part of the core, applying a conservative approach.

2. The ΔR may be consistent across time at this location, but that does not mean it is consistent across archive. The 600 ΔR value was obtained through a coral study, yet none of the proxies used were corals. For the benthic foraminifera, there is no species listed, yet there could be an expected difference in reservoir age between an infaunal and epifaunal species.

There could also be a difference between those species and Echinoidea, for example. This is a shallow core, so it may be the case that the same reservoir age can be used, but this choice would need to be justified with literature.

Tesi et al. (2020)¹¹ demonstrated that the ΔR is highly consistent across different organisms, including both benthic and planktic foraminifera from various habitats. Furthermore, we leveraged the tephra layer to reduce the uncertainties associated with the reservoir correction (ΔR). The new approach is well aligned with previous ΔR estimates by Hall et al (2010)¹² using corals across the Ross Sea but yields a lower error, which was instrumental in enabling a new spectral analysis that now incorporates age-depth model uncertainties using a completely different methodology.

3. The lowest error on radiocarbon age is at the deepest core depth which is highly unusual.

We rechecked the data provided by the lab. The lower error is likely due to the amount of material used, which typically helps reduce uncertainty. Overall, the errors are quite similar.

As a suggestion, Mezgec 2017 has a sea ice (Na^+) record, could this be plotted in some way with the IPSO25 to support its use as sea ice proxy?

This is indeed a valuable suggestion. However, with our IPSO₂₅ record, now further supported by new sediment trap evidence, we have demonstrated that our fast ice proxy performs with high reliability. This fits well with the outcomes by Belt et al (2016)¹³ who first described this proxy.

It is also important to highlight that the two proxies are likely capturing different aspects of the sea ice system. Our data specifically reflect fast ice dynamics, whereas the Na^+ proxy is more indicative of large-scale sea ice variability in the Ross Sea or its source region. While fast ice and pack ice along the Victoria Land coast are interconnected, as discussed in our manuscript, the large-scale information provided by the Na^+ proxy and the localized fast ice data from our inlet represent distinct components of the sea ice system.

For these reasons, and to avoid introducing additional uncertainties, we chose not to include the Na^+ record. We are confident that our new data robustly support the reliability of the IPSO₂₅ proxy. Furthermore, IPSO₂₅ has been analyzed in only a limited number of samples, which does not allow for a full reconstruction, whereas the image analysis provides high-resolution down-core data

There should be more exploration into mechanistically how the solar cycle is propagated through the climate system to translate into sea ice extent

In regards to Figure 6, the actual difference in TSI between a max and min of a Gleissberg cycle is 0.17 W/m². This model is forced by a change of ~ 17 W/m² which is orders of magnitude

larger. That allows for much more of a direct effect of shortwave radiation on the region, rather than signal amplification and transport through circulation as most other solar cycle studies would suggest. Instead, there should be more focus on figuring out why such a small direct signal would have the effect it does on inlet opening.

How do these results compare to models that look at changing TSI such as Swingedouw et al., 2010 which found that Antarctic temperatures are expected to cool during peak TSI in solar cycles?

This is good point raised here. In the revised paper, we explicitly acknowledge the limitations of the unrealistic solar forcing used in the experiments. Therefore, the outcomes should be interpreted solely as a means to test hypotheses and to explore the response of key climate-driving elements in Antarctica. In response to this comment, we now provide a more refined interpretation of the relationship between Total Solar Irradiance (TSI) and fast ice in our study region. Specifically, we focus on the role of zonal winds, drawing on evidence from reanalysis datasets and independent ice core records. Additionally, CMIP6 experiments offer further insights, supporting the reanalysis findings and contributing additional mechanistic understanding. The revised documents now include expanded information on the TSI–fast ice relationship, with additional evidence from reanalyses, ice core records, and CMIP6 experiments to strengthen and clarify our interpretation.

We were unable to locate Swingedouw et al., 2010, but we assume the reference pertains to the increase in upwelling associated with the southward migration of the Southern Hemisphere Westerly Winds (SWW), which would be expected to promote enhanced intrusion of modified Circumpolar Deep Water (mCDW) onto the continental margin.

If the takeaway from the paper is that changing TSI associated with solar cycles influences fast-ice extent, then the portion about IPO and AMO should be shortened/de-emphasized. It can be mentioned that the 30-year signal's presence could be the result of those processes to explain its presence, but expanding upon them diverts from the paper's purpose.

Why is the Gleissberg cycle so prominent in the fast ice record if it is so weak in the Steinhilber TSI record where the 200 year cycle is stronger? As a side question: in creating the spectra, was the entire Steinhilber record used or just the time interval overlapping with TR-17-08?

With the new analyses, as previously explained in response to Reviewer 1, the high-frequency signal is smoothed out suggesting this was also affected by noisy signal. Given all the new information and for the sake of simplicity, we have removed the comparison with the Steinhilber TSI record, which displays a complex pattern, likely due to methodological differences, falling outside the scope of this paper. Most importantly, the low-frequency cycles remain prominent, regardless of the analytical approach used.

Minor General Comments:

1. In explaining the importance of fast-ice, I would be interested to see how the fast ice and total sea ice extent are related. What is the evidence that IPSO25 is fast-ice specific?

The evidence provided by Belt et al. (2016)¹³ is strong, and we now also include sediment trap data that further support the exceptional reliability of IPSO25 as a fast ice proxy.

2. Figure 3: It would look a lot cleaner and convey the cycles better to only show the moving average. There is so much noise. That way the y-axis can be constrained and the trends will be more visible.

The moving average is shown later on in the new image showing the spectral analysis outcomes

3. Figure 4: The axis colors are very confusing. Why are the highest and lowest values represented by the same color? It makes it unclear if the light lamina have very high or very low *C. pennatum* biovolume.

Export issue during the pdf creation, the original figure is fine.

4. Figure 1d- The labels are much too small

This Figure has been completely replaced

I am excited to see the next iteration of this paper and what the authors are able to find!

Reviewer #3 (Remarks to the Author):

The manuscript by Tesi and colleagues presents a reconstruction of fast ice in the northwestern Ross Sea, Antarctica, over the past 3,700 years. Most previous studies have focused on pack ice, or simply sea ice, as distinguishing between pack and fast ice dynamics during the Holocene and glacial–interglacial timescales is particularly challenging. Consequently, much less is known about past fast ice dynamics, despite its significant influence on coastal ecosystems, marine-terminating glaciers, and ocean circulation. Placing current and future changes within a natural context is therefore highly commendable. The manuscript is well-written and well-illustrated, with clear objectives and sufficient background information. However, I have several important

concerns regarding the age model and modelling approach, which must be addressed before further consideration. I hope these comments will contribute to improving what is undoubtedly an interesting study.

Major Concerns and Recommendations

Representativeness of the Study Area

The Edisto Inlet is a remote fjord that is partially isolated from global ocean circulation. While I do not question that the proposed proxies enable the reconstruction of fast ice dynamics within this inlet, I am concerned that the reconstructions may be highly site-specific, reflecting a regional rather than a broader response to global forcing due to the physiographic setting (i.e., a small fjord with two marine-terminating glaciers and no clear evidence of modified Circumpolar Deep Water (mCDW) intrusion).

One example is the differing interpretations of the dark and light laminae preserved in the studied core. In this study, dark laminae are interpreted as indicative of early summer sea ice melting (associated with sea ice diatoms and terrigenous supply), whereas light laminae are associated with summer ice-free conditions (linked to open ocean diatom production and rapid settling). However, this interpretation is opposite to findings from Wilkes Land, where dark laminae were deposited during the summer open ocean season, reflecting lower productivity and hyperpycnal plume input (Maddison et al., 2006, 2012; Denis et al., 2012). Although the laminae appear to be well interpreted in this study based on discrete sampling, the use of thin sections would undoubtedly provide further validation.

Additionally, there is no mention of the potential influence of the El Niño–Southern Oscillation (ENSO), which has been shown to exert a strong control on coastal conditions at multi-decadal timescales throughout the Holocene (Shevenell et al., 2011; Pike et al., 2013; Crosta et al., 2020; Johnson et al., 2021, and references therein).

Overall, I recommend strengthening the justification for the significance of this region, study, and interpretation to ensure greater alignment with the broad readership of Nature Communications.

Our MODIS results demonstrate that pack ice dynamics in the region near Victoria Land are closely linked to fast ice behavior, indicating that conditions within the inlet are influenced not only by local factors but also by broader regional processes. In terms of proxy interpretation, the sediment trap results provide valuable insight, particularly highlighting the utility of IPSO_{25} and $\delta^{13}\text{C}$ as reliable indicators, which further supports our initial hypothesis and is also corroborated by taxonomic evidence.

As shown by Holland et al. (2017)⁵ using reanalysis data, summer pack ice anomalies in the Victoria Land region, critical for the opening of the inlets, are associated with zonal wind anomalies over the southern Pacific. Taken together, these findings strongly suggest that the opening of the northern Victoria Land fjords is primarily driven by regional climate dynamics rather than isolated local phenomena.

Regarding ENSO, its frequency could be too high to be reliably detected in our record. This limitation is compounded by the new spectral approach, which incorporates age uncertainties and tends to smooth out high-frequency signals, thereby reducing both noise and short-lived climate events. This does not mean that an ENSO signal is definitively present but undetectable; rather, it suggests the possibility that such signals may be partially smoothed out. In principle, a longer record would allow for better resolution of shorter cycles with greater clarity.

Nonetheless, the strength of this study lies in its ability to resolve long-term climate cycles, offering valuable insights into a less explored aspect of natural climate variability.

Age Model and Its Impact on Results and Interpretations

The age model has been developed using a state-of-the-art Bayesian approach applied to radiocarbon dates obtained from carbonate material, which is commendable. However, several aspects of the age model warrant further discussion.

The age-depth relationship appears to be constant between radiocarbon dates, which does not align with the highly laminated sediment structure. The authors state that the sharp, bright layers represent seasonal, short-lived events, whereas the more diffuse dark layers correspond to multiple years of accumulation. This suggests that sedimentation rates vary considerably between bright and dark layers, which would significantly impact the age distribution between control points. For instance, bright laminae can be several centimetres thick (Figure 2), representing just a few weeks of sedimentation, whereas dark laminae may correspond to several years of accumulation.

Additionally, it is unclear how years with no sedimentation, when sea ice did not open (lines 119–120), have been accounted for in the model.

I was also unable to find any information on how the core-top was dated. Was the sediment-water interface preserved, and was it set at the year of collection? Clarifying this would help assess the accuracy of the most recent age constraints.

Moreover, the uncertainty in the age model increases with depth, ranging from ± 50 to ± 300 years, which represents up to 10% of the total time span covered by the core. Given the floating ages between radiocarbon dates due to highly variable sedimentation rates, as well as the inherent uncertainties of radiocarbon dating, I question the robustness of the age assignment for individual laminae. This, in turn, raises concerns about the reliability of the spectral analyses

and their interpretations, particularly regarding the identification of solar cycles.

I recommend further discussion on these points to reinforce confidence in the chronology and its implications for the study's key findings.

Regarding the core top, as explained in our response to Reviewer 2, we conducted ^{210}Pb analyses and confirmed that the core is indeed missing its uppermost section. To be conservative, we constructed the age-depth model using only the interval between the top tephra layer and the lowermost radiocarbon date. Additionally, we refined the ΔR uncertainties (see further details in the main text), which allowed for improved spectral analysis that now explicitly incorporates age uncertainties.

To address potential age gaps, we also performed ^{210}Pb analysis on a multicorer sample collected at the same site. The results indicate that, despite the laminated sediment structure, the sediment accumulation rate has remained relatively constant over the past century. This does not preclude year-to-year variability in sedimentation, indeed, such variability is likely, but the laminations do not suggest discontinuous deposition at decadal to centennial timescales. This interpretation is further supported by two independent age-depth models developed using ^{210}Pb and radiocarbon data.

Spectral Analyses: Approach and Interpretations

The use of REDFIT, a stationary approach applied to unevenly spaced samples, has identified significant cyclicities (~ 90 years and ~ 210 years) above the 95% confidence interval. These periodicities align with similar cycles observed in the Total Solar Irradiance (TSI) record. However, the reliability of the spectral analysis is highly dependent on the robustness of the age model, which, as outlined in my previous comments, remains questionable due to uncertainties in sedimentation rates and floating radiocarbon ages.

Given these concerns, I recommend incorporating a non-stationary approach to assess how these cyclicities evolved over the considered period. This would provide a more nuanced interpretation of the temporal variability in climate-forcing mechanisms and strengthen the study's conclusions.

In response to this suggestion, we have now included a spectral analysis that takes age uncertainties into account. This improves the reliability of our results by reducing the risk of misinterpreting periodic signals due to dating inaccuracies. By incorporating these uncertainties, we can better identify long-term climate cycles. Further details on the method can be found in the main text. This approach, however, tends to smooth out high frequency cycles as explained above. we updated our methodology by implementing a Monte Carlo-based sensitivity analysis. To do so, this new approach involved generating 1,000 possible age-depth realizations and calculating the power spectrum for each one. To evaluate the stability of the

spectral features, we compared the results against a red-noise background using the method outlined by Schulz and Mudelsee (2002)¹. This more robust, statistically grounded approach confirms the persistence of two distinct low-frequency cycles, in agreement with our initial results.

Solar Irradiance and Model Output

The manuscript states in Methods (Section 4.6) and the Supplementary Information (Section 4) that pre-industrial (PI) conditions were perturbed with an abrupt 4% increase in the solar constant lasting 150 years. However, this perturbation appears to be well outside the natural range of variability, raising concerns about the robustness of the model output and its associated interpretations.

Magnitude of the Perturbation: It is unclear what the 4% increase refers to. If it applies to Total Solar Irradiance (TSI), this would correspond to an increase of approximately 50 W/m² from the mean PI TSI value of ~1364 W/m². This far exceeds known variations in the solar constant, which are approximately 1 W/m² for the 11-year Schwabe cycle, and 4–5 W/m² for the 200-year Suess cycle (Dean, 2000; Schmutz, 2021).

For context, the reduction in TSI during the Maunder Minimum was only ~0.2%, and current research suggests that the Little Ice Age (LIA) was primarily driven by volcanic eruptions rather than changes in solar irradiance (McGregor et al., 2015). Changes in solar irradiance over the Holocene have also been relatively minor, reaching only a few W/m² (Vieira et al., 2011).

Forcing Duration and Natural Cyclicity: The application of a 150-year constant forcing is inconsistent with natural solar cycles. For example, if considering the Suess cycle, irradiance varies from minimum (~1364 W/m²) to maximum (~1366 W/m²) over ~100 years. The imposed perturbation thus lacks a realistic temporal evolution.

As explained for reviewer#2, we acknowledge that a 4% increase in solar forcing represents an idealized and physically unrealistic scenario, and that a more moderate forcing would likely produce a weaker surface climate response. In the revised manuscript, we have clarified that the Abrupt-Solp4p experiment is intended to investigate the potential sensitivity of the climate system to solar forcing, rather than to replicate past or projected climate conditions. In the revised version, we have made clear and emphasized that the applied solar forcing in this simulation far exceeds natural variability; thus, the results should be interpreted as providing mechanistic understanding rather than direct climatic evidence, serving to complement insights from more realistic studies.

Model Response to Forcing: It is also unclear why a constant 4% increase in the solar constant resulted in a continuous increase in surface shortwave (SW) radiation over the modelled 150 years. The magnitude of this increase ($\sim 20 \text{ W/m}^2$, Figure 6a) is substantial compared to changes in astronomical insolation. For instance, summer insolation at 60°S increased by only $\sim 10 \text{ W/m}^2$ throughout the entire Holocene (Berger & Loutre, 1991). This excessive forcing likely explains the significant rise in sea surface temperature (SST) by 1.5°C in the Ross Sea region (Figure 6a). A more reasonable and geophysically realistic perturbation would have resulted in a more tempered response.

As said before, we agree that a 4% increase in solar forcing represents an idealized and geophysically unrealistic perturbation, and that a moderate forcing would likely result in a weaker surface response. We have clarified in the revised manuscript that the Abrupt-Solp4p experiment is used in this study to explore the potential climate system response to solar forcing, rather than to reproduce past or projected realistic climate conditions. We have added a cautionary note in the discussion section further clarifying this.

Despite a constant 4% increase in solar forcing, there is a continuous rise in surface temperature, which can be attributed to internal feedbacks within the coupled ocean-atmosphere-ice system, including ice-albedo feedbacks, ocean-atmosphere heat exchange, the cloud radiative feedbacks evolve over time continuously amplifying surface response, as suggested by papers like Goosee et al (2018) and Aereson et al (2024)^{14,15}

In the Abrupt-Solp4p experiment, increased insolation directly drives sea-ice retreat in spring and summer (Zhang et al., 2010)¹⁶. Additionally, strengthened westerlies in response to increased solar forcing promote export and dynamic thinning of sea ice in the western Ross Sea, further accelerating sea ice melt (Holland et al., 2017)⁵. This reduction in sea decreases surface albedo and increases the absorption of shortwave radiation resulting in surface warming. The loss of sea ice also promotes atmosphere-to-ocean heat transfer, delaying the formation of sea ice in autumn and winter.

Moreover, the increased solar constant may influence cloud properties, further contributing to surface warming. For example, comparing the final 30-year mean cloud results from the Solp4p experiment with pre-industrial simulations, Aerenon et al. (2003, 2024)^{15,17} reported an increase in optically thick low- and mid-level clouds poleward of $\sim 60\text{S}$. These cloud adjustments significantly contribute to the total cloud radiative effect, even after 150 years of simulation, by trapping the outgoing longwave radiation thereby causing surface warming. The combine influence of these processes causes a continuous rise in SST despite the constant 4% solar forcing increase in the Abrupt-Solp4p experiment.

Implications for Fast-Ice Dynamics: This disproportionate forcing may also contribute to the apparent contradiction between modern observations and the model output. Specifically, in situ observations suggest that atmospheric temperature plays only a minor role in fast-ice melt across Antarctica, whereas mechanical breakup—driven by wind and waves—is the dominant factor, particularly when pack ice no longer provides protection (Lines 241–244). The model, however, appears to attribute significant fast-ice loss to atmospheric warming, possibly due to the exaggerated solar forcing.

In the new version, we have moderated the interpretation of the model outcomes, although we still consider them valuable for offering insights that complement findings from the literature, particularly reanalysis-based studies. We fully acknowledge the differences between the model results and observational evidence. Nonetheless, the model highlights changes in zonal winds, a factor that has been relatively overlooked in the previous version, which we believe deserves attention. While the model also shows contributions from increased radiation and associated warming, we agree that these effects may be overstated due to the idealized forcing. That said, previous studies have demonstrated that solar radiation, especially in the Ross Sea region, can play a significant role in sea ice loss, suggesting that the mechanisms identified in the model may still have relevance under certain conditions.

Model Resolution and Applicability to Edisto Inlet: Given that Edisto Inlet is a narrow fjord, it is important to confirm whether the model resolution is sufficient to accurately capture local processes. If the spatial resolution is too coarse, key dynamics—such as interactions between fast ice, ocean currents, and atmospheric forcing—may not be well resolved, leading to potential misinterpretations of the model results.

We agree that model resolution is a critical limitation when interpreting results for narrow fjord systems such as Edisto Inlet. The global climate model used in the CMIP6 framework lacks the spatial resolution needed to resolve small-scale processes such as local ocean circulation, wind-channeling effects, and fast ice–pack ice interactions specific to the inlet. In fact, the CMIP do not model fast ice at all, so we do not use the model to draw conclusions at the fjord scale. Instead, it provides a broader regional context, including sea ice in the Ross Sea, complementing high-resolution observations and reanalyses.

Recommendations: (1) Clarify what the 4% increase in solar constant refers to and ensure it falls within geophysically plausible values; (2) Consider using a more realistic time-dependent solar forcing aligned with known solar cycles; (3) Provide further discussion on why the imposed forcing results in a continuous increase in SW radiation and whether this is an artefact of model

setup; (4) Address the potential contradiction between modern observations and modelled fast-ice dynamics; (5) Ensure that model resolution is adequate for capturing processes within the narrow fjord setting of Edisto Inlet.

These refinements would help improve the credibility of the model results and their broader implications for Antarctic fast-ice variability.

We hope that the revisions and clarifications provided above successfully address all the points raised in your comment, and we have updated the manuscript accordingly to reflect these improvements.

Minor concerns and recommendations

References should be cited at their appropriate places within a sentence, rather than grouped together at the end. For example, references [4–10] in lines 54–56 should be assigned to their respective processes. Please ensure this is done consistently throughout the manuscript.

While this approach can be applied to some references, others share similarities, making it difficult to follow the principle consistently. In certain cases, references are placed at the end of the sentence to improve readability. This also depends on writing habits and style, especially since the journal does not explicitly require a specific format. However, we have followed this approach where possible.

The method for sampling paired laminae remains unclear, both in this study and in Tesi et al. (2021). Specifically : How were the paired laminae selected? What was their thickness? Were sub-laminae present, and if so, how were they accounted for? A clearer explanation of the sampling strategy would enhance the reproducibility of the study.

As stated in the text, Tesi et al. (2020)¹¹ selected laminae that were thick enough to be sampled without cross-contamination. This was feasible in Section III of core HLF17-01, but it covered only a limited portion of the core and was therefore time-restricted. In the present study, we applied the same criterion, sampling only laminae thick enough to avoid cross-contamination, but extended it by selecting a pair of laminae from each core section to assess consistency throughout the entire core.

The relevance of diatom biovolume for reconstructing sea ice vs open ocean conditions is questionable. While biovolume is important from physiological and geochemical perspectives, absolute or relative species abundances are more relevant for assessing past environmental

conditions.

Additionally, calculating biovolume introduces significant errors if diatom biometry was not performed. The size of specimens within a single sample, as well as between samples, can vary considerably. For example, *Fragilariopsis curta* frustules range from 10 to 55 μm in a single sample, with mean lengths varying between 21 and 27 μm over the Holocene (Crosta, 2009). The size variations are likely even more pronounced for the tubular *Corethron pennatum*. If biovolume estimates were based on the assumption that larger valves correspond to longer tubes with more girdle bands, it is important to acknowledge whether this relationship has been directly validated.

Given these uncertainties, it may be more appropriate to rely on species abundances rather than biovolume for environmental reconstructions.

With this approach, our intention was not to disregard the established use of relative species abundances in the literature. However, given the structure of the laminated sediments—densely packed with diatoms arranged like microscopic building blocks within each lamina—we believe that biovolume offers a more representative measure of laminae composition. Tesi et al. (2021)¹¹ presented both relative abundance and biovolume data, showing that biovolume can more clearly distinguish differences between laminae. Similarly, Alley et al. observed limited contrasts in laminated sediments until the data were normalized by biovolume, further supporting its usefulness in this context.

While we acknowledge the uncertainties associated with estimating diatom biovolume, particularly due to species-specific morphological variability, these uncertainties are applied uniformly across both dark and light laminae. As such, we maintain that biovolume remains a valid and informative metric for distinguishing between contrasting lamina types in our dataset.

The environmental interpretation of the light and dark laminae in this study differs from other coastal areas, where light laminae typically represent spring melting, and dark laminae indicate summer ice-free conditions with reduced biogenic sedimentation and increased terrigenous input from hyperpycnal plumes. This contrast should be explicitly mentioned in the manuscript. Additionally, a thin-section analysis would be highly beneficial for verifying the interpretation of the laminae structure and depositional processes.

We believe that the paper already presents a substantial amount of diverse data, and therefore we do not find it necessary to further expand the analysis with thin-section work—though we agree that such data would undoubtedly provide valuable additional insights. To address the context of laminae formation, we have added a section in the introduction that further highlights the nature and variability of laminated sediments in Antarctica.

In accordance with international taxonomic rules, genus names must be fully spelled out when starting a sentence (e.g., Line 141). Please ensure this is corrected throughout the manuscript.

We have revised this throughout the text

Lines 161-163. The current wording suggests a strict seasonal signal in the laminae, which contradicts earlier statements. A clearer approach would be to describe. Dark laminae as indicative of years with extensive sea ice cover. Light laminae as deposited occasionally and rapidly during years when the inlet opens up. This explanation is better articulated in other sections of the manuscript and should be consistently applied.

Not exactly, this reflects the brightness values obtained through image analysis. The observed brightness patterns and laminae detection indicate that the number of laminae is significantly lower than the number of years represented in the core, suggesting that the laminae do not reflect a strictly seasonal signal.

The band-pass filter analysis does not seem to add significant value, as it naturally aligns with the original grey-scale record given the spectral analysis results (90- and 210-year cyclicities). The more valuable insight comes from identifying offsets between the raw and filtered records, indicating non-stationarity in the cyclicities. Instead of a band-pass filter, a wavelet analysis would be more appropriate for capturing temporal variations in these cycles.

We believe that including both the bandpass filter and wavelet analysis would not significantly enhance the interpretation, as the bandpass filter already captures the key features of signal amplitude over time, including periods of high and low signal.

Lines 211-212. The sentence regarding the impact of positive SAM (SAM+) is unclear. Generally, a positive SAM strengthens Ekman transport northward, leading to increased sea ice extent and decreased coastal concentration. This seems to contradict the statement in the manuscript. Or are you suggesting that SAM+ enhances sea ice transport from the southern Ross Sea and accumulates it around Cape Hallett? If so, this should be explicitly stated.

As the SAM strengthens, northward sea ice expansion is favored by Ekman transport (Hall and Visbeck, 2002)¹⁸. This process is particularly active during sea ice formation and expansion in the fall and winter. However, Holland et al. (2017)⁵ demonstrated that zonal wind anomalies in the spring promote sea ice export in the Northern Victoria Land region. In fact, zonal wind anomalies (strengthening) over the Pacific are a strong predictor of sea ice opening in northern-western Ross sea area. Thus, zonal winds can play contrasting roles depending on the season.

The section Lines 236-260 is excessively long and does not contribute significantly to the study's core findings. A more concise version would improve readability and focus.

This section has been shorten it

Sections 4.6 in the Methods and section 4 in the SOM are nearly identical, except for Table S2. Either consolidate them or clarify their distinct purposes to avoid unnecessary repetition

We removed that part from the supplementary material providing only the table.

Figure 2: Some sections (e.g., I, VIII, XII) appear massive, lacking clear lamination. Could this be an issue with the photographs, or is it representative of the actual sediment structure? If it is real, a brief explanation of why these sections appear massive would be useful.

To the naked eye, yes; however, the image analysis revealed significantly more laminae than could be detected through visual inspection, offering a more objective and unbiased method for laminae counting. Nevertheless, even with this enhanced approach, the total number of laminae remains well below the number of years represented by the absolute age of the core, as previously discussed.

Figure 4: The current colour scheme is confusing, as yellowish tones appear at both ends of the spectrum. This makes it difficult to distinguish between low and high values. Please replace the very low *Corethron pennatum* biovolume category with purple to create a more intuitive gradient.

This was noticed by all reviewers and it was a problem during the pdf creation.

Lines 47-48 of the SOM: The presence of laminae with distinct depositional timeframes (dark = multi-year accumulation, bright = short episodic events) contradicts the statement that "accumulation rates vary smoothly downcore." Please rephrase this section to reflect the variability in sedimentation rates due to the laminated structure.

As explained in the text and in couple of part of this rebuttal letter, the ^{210}Pb data suggest that despite the laminate pattern, the SAR did not show evidence of rapid change in sedimentation or evident hiatus.

Table S1: The age of the core top is missing. Please report this value and explain how it was determined. Specifically, was the sediment-water interface preserved during core retrieval?

What method was used for age assignment (e.g., ^{210}Pb dating, radiocarbon dating, or another approach)?

Please see detailed comments above. As previously explained, we adopted a conservative approach in constructing the new age model, limiting it to the interval between the top tephra layer and the lowermost radiocarbon date, as the core is missing its uppermost section, confirmed by lack of excess ^{210}Pb .

I hope these comments and recommendation will prove useful.

Cordially.

Xavier Crosta

Reviewer #4 (Remarks to the Author):

I co-reviewed this manuscript with one of the reviewers who provided the listed reports.

This is part of the Nature Communications initiative to facilitate training in peer review and to provide appropriate recognition for Early Career Researchers who co-review manuscripts.

Lines 355-356. The phrase "one distinct specimen" is ambiguous—does this refer to one frustule, or one whole diatom, which consists of two frustules? If the latter was not accounted for, the abundance of *Corethron pennatum* may have been overestimated by a factor of two. Please clarify.

We have clarified this in the text.¹ Schulz, M. & Mudelsee, M. REDFIT: estimating red-noise spectra directly from unevenly spaced paleoclimatic time series. *Computers & Geosciences* **28**, 421–426 (2002).

2. Ushio, S. Factors affecting fast-ice break-up frequency in Lützow-Holm Bay, Antarctica. *Ann. Glaciol.* **44**, 177–182 (2006).
3. Massom, R. A. *et al.* Snow on Antarctic sea ice. *Reviews of Geophysics* **39**, 413–445 (2001).
4. Sturm, M. & Massom, R. A. Snow in the sea ice system: friend or foe? in *Sea Ice* (ed. Thomas, D. N.) 65–109 (Wiley, 2017). doi:10.1002/9781118778371.ch3.
5. Holland, M. M., Landrum, L., Raphael, M. & Stammerjohn, S. Springtime winds drive Ross Sea ice variability and change in the following autumn. *Nat Commun* **8**, 731 (2017).
6. Stewart, C. L., Christoffersen, P., Nicholls, K. W., Williams, M. J. M. & Dowdeswell, J. A. Basal melting of Ross Ice Shelf from solar heat absorption in an ice-front polynya. *Nat. Geosci.* **12**, 435–440 (2019).
7. Munro, D. R., Dunbar, R. B., Mucciarone, D. A., Arrigo, K. R. & Long, M. C. Stable isotope composition of dissolved inorganic carbon and particulate organic carbon in sea ice from the Ross Sea, Antarctica. *J. Geophys. Res.* **115**, 2009JC005661 (2010).
8. Mayewski, P. A. *et al.* Ice core and climate reanalysis analogs to predict Antarctic and Southern Hemisphere climate changes. *Quaternary Science Reviews* **155**, 50–66 (2017).

9. Bertler, N. A. N. *et al.* Solar forcing recorded by aerosol concentrations in coastal. *Ann. Glaciol.* **41**, 52–56 (2005).
10. Mayewski, P. A. *et al.* Solar forcing of the polar atmosphere. *Ann. Glaciol.* **41**, 147–154 (2005).
11. Tesi, T. *et al.* Resolving sea ice dynamics in the north-western Ross Sea during the last 2.6 ka: From seasonal to millennial timescales. *Quaternary Science Reviews* **237**, (2020).
12. Hall, B. L., Henderson, G. M., Baroni, C. & Kellogg, T. B. Constant Holocene Southern-Ocean 14C reservoir ages and ice-shelf flow rates. *Earth and Planetary Science Letters* **296**, 115–123 (2010).
13. Belt, S. T. *et al.* Source identification and distribution reveals the potential of the geochemical Antarctic sea ice proxy IPSO25. *Nat Commun* **7**, 12655 (2016).
14. Goose, H. *et al.* Quantifying climate feedbacks in polar regions. *Nat Commun* **9**, 1919 (2018).
15. Aeronson, T., Marchand, R. & Zhou, C. Cloud Responses to Abrupt Solar and CO₂ Forcing: 2. Adjustment to Forcing in Coupled Models. *JGR Atmospheres* **129**, e2023JD040297 (2024).
16. Zhang, Q. *et al.* Climate change between the mid and late Holocene in northern high latitudes – Part 2: Model-data comparisons. *Clim. Past* **6**, 609–626 (2010).
17. Aeronson, T. & Marchand, R. Cloud Responses to Abrupt Solar and CO₂ Forcing Part I: Temperature Mediated Cloud Feedbacks. Preprint at <https://doi.org/10.22541/essoar.169945375.53399648/v1> (2023).
18. Hall, A. & Visbeck, M. Synchronous Variability in the Southern Hemisphere Atmosphere, Sea Ice, and Ocean Resulting from the Annular Mode*. *J. Climate* **15**, 3043–3057 (2002).

We sincerely thank the reviewers for their thorough evaluation and constructive comments, which have greatly helped us improve the manuscript. Below, we present the reviewers' comments followed by our detailed responses, provided in *blue italics* for clarity.

Reviewer #1 (Remarks to the Author):

General comments:

We appreciate the authors' attention to detail and consideration of all the comments from our initial review. This version represents a significant improvement. The figure quality is much improved and this improves the clarity of the manuscript.

Major Comments:

Line 520: The sea ice data analysis methods section, Section 4.7, does not include how landfast ice within Edisto Inlet was defined and differentiated from mobile sea ice within the inlet. If the authors are defining all pixels which were identified as ice, based on the ice surface temperature, within Edisto Inlet as landfast ice we do not find this convincing enough evidence to be called landfast and not just sea ice or ice covered. To be defined as landfast there needs to be a component of immobility over a period of time in addition to being identified as sea ice. Or is it based on the diatom assemblages commonly associated with fast ice? At a minimum, this caveat should be acknowledged in the manuscript.

We thank the reviewer for this valuable comment. We have clarified in the Supplementary Information how landfast ice within Edisto Inlet was defined and differentiated from mobile sea ice. Specifically, we examined satellite visible imagery for each year, which consistently shows that the sea ice within Edisto Inlet behaves as persistent landfast ice, remaining anchored to the coast rather than drifting with winds and currents as the pack ice outside the bay does. This interpretation is further supported by repeated in situ summer observations collected by Italian research expeditions over the past decade, which confirm the immobility of the ice within the inlet.

We have incorporated this explanation into the Supplementary Information, including illustrative examples, and have added a note in the Methods section (Section 4.7) acknowledging this definition and clarifying the observational basis used to distinguish landfast ice from mobile pack ice.

Minor Comments:

Line 75-77: the sentence ... with East Antarctic and Bellingshausen Sea exhibiting the largest positive trend (about 1.1% yr⁻¹)... Victoria and Oates land (-1.8% yr⁻¹) is not entirely accurate and slightly misleading. From Fraser et al. (2021) 1) there is no East Antarctica region, but I am assuming the authors meant the Australia region. 2) by the percentages the region with the largest positive trend is the Bellingshausen Sea (2.8 %yr⁻¹) and largest negative trend Weddell Sea (~-2.6 %yr⁻¹). If you are saying a region is the largest and then use percentages, the regions with the largest percent should be used.

We thank the reviewer and we have edited the text accordingly, using highest and lowest values observed

Line 58: "bottom waters" to "bottom water"

Changed

Line 114: This should refer to Figure 1a not just Figure 1

Changed

Line 126: Instead of referring to the “Methods” section this sentence should refer to the section number (i.e. Section 4.2 for the age depth model).

Corrected

Line 215: a reference to Figure 1a and the yellow polygon denoting the region of northern Victoria Land where the pack ice concentration was calculated would improve the clarity

Changed accordingly

Line 233: From Figure 7 there seems to be a lagged response of ~15 days from when the pack ice concentration decreases to the ice-free inlet % going up. This is speculative and based on the visuals of Figure 7. However, if there is a consistent lagged relationship this would provide grounds for some speculation and discussion on the physical link between pack ice concentration and the ice-free Edisto Inlet.

This is an interesting point, which we also addressed in the manuscript to some extent, where we note that 'pack ice acts as a barrier, reducing exposure to solar radiation, surface winds, and ocean waves.' I believe we can account for all these processes, which are listed as potential drivers, to explain the link between pack ice and fast ice inside the fjord. Yet, I do not think we have enough evidence to pinpoint a specific process, unfortunately.

Line 368: This is merely a suggestion and realize that we as the reviewer should not dictate the content of another's paper. Since the main point of this paper is the ability to say if Edisto Inlet was ice free or not for the past ~3.7 thousand years we feel like there is a figure missing that displays a general idea of the trend of ice covered or not over the period of the core. We envision something like a climate stripes figure (examples can be found <https://www.reading.ac.uk/planet/climate-resources/climate-stripes>). For this manuscript the lines could be binary indicating if Edisto Inlet was ice covered or not during each year or each stripe could be a percentage of a 10 year where hues of blue indicate more ice covered years and hues of reds indicate more ice free years for example. The reviewers believe a figure like this would provide visual evidence for the periodicity of ice covered, vs not, throughout the time represented by the core. Without a figure, and accompanying text, displaying the ice conditions of Edisto Inlet over the period of the core we feel a large and important part of this paper is missing.

We followed the suggestion and added another figure with a 5-yr bin (new Figure 6).

Line 536: Solp4p needs to be capitalized.

Corrected

Line 555: Figure 1b. This figure should be more zoomed in on Edisto Inlet such that a reader could differentiate between the 3 points. Cape Hallett should also be annotated on Fig 1b, since it is referred to in the text.

The edited Fig.1 accordingly

Line 659: No need to refer to Figure 1b here. Could be reworded to the percentage of Edisto Inlet ice free.

Corrected

Line 609: Is there a high pass filter line within Figure 3a? If so, it should not be black. If not, our apologies, as the print makes it look like there is a black high pass filter line.

No, there is not BP filter here

Paragraph beginning line 270, plus throughout the discussion: It is hypothesized a few times that solar absorption in an adjacent non ice-covered ocean (e.g., if pack ice has retreated) can weaken fast ice and eventually lead to its melt or breakout. There is actually a good reference for this hypothesis: <https://doi.org/10.5194/tc-14-2775-2020> - which is currently not included in the reference list. I think it could strengthen this argument.

We added the new reference

General for Figures: The way the author denotes which axis corresponds to various lines within a plot changes. Adopting a consistent format is suggested. For example: Figure 5a/b the axis tick colours denote which axis is associated with which line while figure 6b/c the axis label colour denotes that the right axis is associated with the red line. List of figures with inconsistent notation of associated axis: Figure 5a/b, 6b/c, 7a-f, 8d.

We uniformed labels and axes across images

Reviewer #3 (Remarks to the Author):

Tesi and co-authors have provided an insightful rebuttal letter and a thoroughly improved revised version of the manuscript. I commend the considerable amount of work undertaken to address the reviewers' comments. I particularly appreciated the addition of modern data demonstrating that sea ice conditions in the inlet are of regional relevance and that the proxies reflect different aspects of sea ice variability. The revisions have addressed most of my initial concerns; however, two main issues remain.

Firstly, I still believe that the band-pass analysis is not appropriate, as such methods tend to reveal cycles regardless of their significance. The fact that the amplitude of the cycles varies over time suggests they are non-stationary. This is further supported by the ~90-year cycle falling below the 95% confidence interval. I therefore recommend replacing the band-pass and spectral analyses with a wavelet analysis, which

accounts for non-stationarity and would be far more suitable. I raised this point in the first round of review, but did not receive a satisfactory response. The revised spectral analysis, which incorporates age uncertainties, still does not address the non-stationarity of the cycles.

We agree with the Reviewer that the cycles are not stationary, which is common in climate records (e.g. Sorrell et al., 2012; Debret et al., 2009), and we appreciate the suggestion to use wavelet analysis. However, applying a wavelet transform to a single chronology (e.g. the median age model) would be inconsistent with our approach, which explicitly propagates age-model uncertainties into the spectral analysis. A wavelet applied to one realization does not capture the full uncertainty space and may therefore yield misleading results. To our knowledge, no established methods currently exist to incorporate age uncertainties directly into wavelet analysis.

To address the Reviewer's concern, we performed an additional analysis CONSISTENT with our uncertainty-propagation framework. Specifically, we extracted those age-model realizations in which solar spectral peaks (de Vries and Suess cycles) exceed the statistical threshold (see Methods), and then propagated the corresponding age uncertainties into the band-pass filtered greyscale record (new Fig. 7). This analysis highlights the persistence of the solar cycles over time given chronological errors, and emphasizes intervals where the signal is weaker or blurred (either due to said age errors, or a weaker solar-proxy link). The results are consistent with our original findings (Fig. 6), but more clearly show the variability in signal amplitude over time. To our knowledge, this is the first time such an approach has been applied, and we consider it an effective and more transparent alternative to overcome the limitations of wavelet analysis.

As to the comment regarding the ~90-year cycle, we believe there may be a misunderstanding. The Reviewer notes that this cycle falls below the 95% confidence interval, but we would like to stress that in our ensemble-based approach significance is evaluated across 1,000 age-model realizations. Our results show that spectral peaks near 90 years exceed the confidence threshold in more than 5% of the age model realizations, potentially up to ~50%. Thus, while the 90-year spectral peak does not consistently exceed the significance threshold across all realizations (and as such we treat it cautiously in the discussion), our ensemble analysis shows that it is present in a large portion of the possible chronologies. This is also why we express the significance threshold using a conservative 5th percentile across the ensemble, i.e. accepting that any solar signals may be blurred by age uncertainty.

Secondly, I remain concerned about the discrepancy between the primary interpretations drawn from the data—highlighting winds as the main driver of sea-ice variability in the northern Ross Sea and the inlet—and those based on the modelling results, which suggest temperature as the dominant factor. This contradiction is only briefly acknowledged in the main text. The imposed 4% increase in solar forcing is clearly beyond observed TSI variability, and this should be explicitly stated. I frequently noted that the interpretations derived from the model contrast with those based on the empirical data (see the annotated file). The added value of the model must be more clearly articulated and substantiated. I also question—though this is outside my area of expertise—whether more climatically relevant model experiments exist, for example those using more realistic solar forcing scenarios.

We appreciate the reviewer's thoughtful comment. We believe that the limitations and strengths of the CMIP6 experiments have been properly acknowledged in the revised manuscript. In particular, at line 350 we explicitly state: "It is important to note that the solar forcing applied in these simulations (4%) significantly exceeds natural variability; therefore, these results should be interpreted as offering mechanistic insights rather than direct evidence, complementing findings from re-analyses."

The main advantage of the CMIP6 experiments lies in providing an additional, model-based perspective on the processes driving sea-ice variability linked to solar variability. To our knowledge, this is the only multi-

model ensemble of Earth System Models (ESMs) that has performed such a multi-model experiment specifically addressing solar influence. Other simplified modelling studies exploring solar forcing have produced complementary insights into climate responses, and these are cited in the manuscript to provide context and demonstrate the current state of knowledge.

First of all, Holland et al. (2017) provides a strong (convincing mechanistic driver) foundation for our study, offering clear evidence that wind-driven sea-ice export is a key mechanism in the region. In particular, given the tight coupling between sea ice inside and outside the fjord, it is plausible that zonal wind anomalies are responsible for the open-water conditions observed within the fjord. While no existing models are specifically designed to reproduce these local wind–ice interactions under varying solar forcing scenarios in our region, the CMIP6 experiments provide valuable complementary insight by highlighting large-scale circulation changes AND associated thermodynamic responses .

In addition, we have included further evidence from the literature addressing the interactions between solar variability and wind anomalies. Collectively, these studies demonstrate that the relationship is robust across different modelling approaches, including the Solp4p experiment. Importantly, this experiment also highlights additional processes, such as temperature/heat responses, which are particularly relevant when considering the combined effects of MULTIPLE mechanisms operating on sea ice. For example, rapid sea-ice export can expose the ocean surface, allowing enhanced absorption of solar radiation and promoting melt well documented in the Ross Sea (e.g., <https://www.nature.com/articles/s41561-019-0356-0>). Therefore, our study does not advocate a SIGNLE dominant process, but rather adopts an INTEGRATIVE perspective that encompasses both dynamic and thermodynamic drivers and their interactions.

In brief, what may appear as inconsistencies between the observational and modelling results should instead be viewed as evidence derived from different but complementary approaches. Together, these perspectives strengthen rather than contradict each other, contributing to a more comprehensive understanding of the coupled processes governing sea-ice behavior in the region.

We have modified the last part of the discussion (line 367-369) conclusion to further highlight the integrative perspective that we are trying to convey, emphasizing the complementary roles of winds, ocean dynamic and thermodynamic processes.

Comments from the annotated file:

I failed to understand how the section above the tephra horizon (upper 55.5 cm) was dated. Using the mean SR?

To be conservative, we did not date the core above the tephra because of the lack of tie points.

From X years BP to 3.7 ka BP

This refers to the basal age which is consistent with previous age depth model. So there is not need to add the range

March 22 to Feb 2023 in Figure 5

Thanks for noticing, this was a typo.

No diatom to support

We thank the reviewer for this comment. We indeed have additional data on diatoms and on open-water biomarkers; however, these results are part of a separate ongoing study. Nonetheless, it is worth noting that these new data fully support the conclusions presented here. We also identified and corrected an error in the formula used to calculate the IPSO fluxes. The revised figure now reflects the corrected values, which continue to support our original interpretations and remain consistent with the concentrations of individual laminae presented in Fig. 4.

I do not get how one can infer mode of deposition (rapid termination vs base sedimentation) from the brightness itself. One situation could be fast termination of consecutive dark layers, with no interbedded light laminae.

This refers to the variability within each class. The white laminae are more homogeneous and exhibit lower internal variability in brightness, suggesting that they represent rapid and discrete depositional events rather than the cumulative outcome of multiple processes.

Threshold values to separate low from high brightness values?

It is not simple as that. The lamina identification was fully explained in the Methods (section 4.5)

Please provide the range of the cycles: 90-110 years (plateau) and 240-280 years (slight plateau and bump). The ~90 year cycle is below the 5% CI? Is it relevant? A wavelet analysis will show if it is present throughout the record. The 90 yr cycle has periods of low amplitude, probably showing non-stationarity.

See previous answer, first point

How do you explain sedimentation when fast ice does not break up and covers the core site? Do you expect high focusing from outside the inlet as observed in other funnel-like (MD03-2601; Denis et al., QSR 2009) or contouritic systems (IODP U1357; Johnson et al., NatGeo 2021)?

After the first year of maintenance, the mooring line was redeployed for a second year. However, the fjord did not open during the summer of 2024, so the instruments were only recovered in March 2025. Most of the sediment trap cups were empty, with only a few containing small amounts of material. This, of course, does not provide information about near-bottom dynamics. We are currently analyzing these samples to better understand the potential sources of material under sea-ice-covered conditions. At this stage, we do not have sufficient evidence to draw conclusions, and any interpretation would be speculative. Nevertheless, both the long core and multicorer data indicate that sediment accumulation is generally continuous, even within this highly dynamic environment.

Isn't it contradictory to Solp4p output?

We believe we extensively answered about this just above (second point)

As there are no vectors, it is difficult to know in which direction blow the zonal winds. North of 70°S => mean the negative anomalies north of Victoria Land? However, Fig 8c shows positive anomalies over

Victoria/Edisto region. Depending on the latitude, northerly winds have been shown to either melt (margin; Turner, GRL 2017) or compact (coast; Sugimoto, Polar Sci 2021). Do you expect melting to overcome compaction here?

We have added an additional figure in the Supplementary Information to better illustrate the CMIP results. This figure shows, in addition to the increase in zonal winds that favour sea ice export, the intrusion of warm air into the Ross Sea. The main text has been modified accordingly so now we refer to this image in the Supplementary Information.

Opposite to Solp4p results

Not exactly. Here we are presenting evidence from studies on fast ice, most of which tend to examine individual processes in isolation. Our interpretation, however, is that sea-ice variability results from the combined action of multiple interacting processes. The initial trigger is likely the state of the pack ice—if this is not removed, then waves, local winds, and heating of the ice-free ocean adjacent to the fjord are unlikely to be as effective. As explained in the text, the Sol4p experiment does show zonal wind anomalies like what reported in Holland et al as well as other thermodynamic processes. Please also refer to our response to the second point for further clarification.

I hope these suggestions will help enhance what is already an engaging and valuable study, likely to be of considerable interest to the broad readership of Nature Communications.

Reviewer #4 (Remarks to the Author):
